# Technical Note: Seamless gas measurements across Land-Ocean Aquatic Continuum - corrections and evaluation of sensor data for $CO_2$, $CH_4$ and $O_2$ from field deployments in contrasting environments

Anna Rose Canning[1,2], Peer Fietzek[3], Gregor Rehder[4], and Arne Körtzinger[1,5]

[1]GEOMAR Helmholtz-Zentrum für Ozeanforschung, Kiel, Schleswig-Holstein, Germany
[2]-4H-JENA engineering GmbH, Jena, Germany (formerly Kongsberg Maritime Contros GmbH, Kiel, Germany)
[3]Kongsberg Maritime GmbH, Hamburg, Germany
[4]Leibniz Institute for Baltic Sea Research Warnemünde, Rostock-Warnemünde, Germany
[5]Christian-Albrechts-Universität zu Kiel, Kiel, Schleswig-Holstein, Germany

**Correspondence:** Anna Rose Canning (acanning@geomar.de)

**Abstract.** The ocean and inland waters are two separate regimes, with concentrations in greenhouse gases differing on orders of magnitude between them. Together they create the Land-Ocean Aquatic Continuum (LOAC), which comprises itself largely of areas with little to no data in regards to understanding the global carbon system. Reasons for this include remote and inaccessible sample locations, often tedious methods that require collection of water samples and subsequent analysis in the lab, as well as the complex interplay of biological, physical and chemical processes. This has led to large inconsistencies, increasing errors and inevitably leading to potentially false upscaling. A set-up of multiple pre-existing oceanographic sensors allowing for highly detailed and accurate measurements was successfully deployed in oceanic to remote inland regions, over extreme concentration ranges. The set-up consists of 4 sensors measuring $pCO_2$, $pCH_4$ (both flow-through, membrane-based non-dispersive infrared (NDIR) or Tunable diode laser absorption spectroscopy (TDLAS) sensors), $O_2$, and a thermosalinograph at high-resolution from the same water source simultaneously. The flexibility of the system allowed for deployment from freshwater to open ocean conditions on varying vessel sizes, where we managed to capture day-night cycles, repeat transects and also delineate small scale variability. Our work demonstrates the need for increased spatiotemporal monitoring, and shows a way to homogenize methods and data streams in the ocean and limnic realms.

## 1 Introduction

Both carbon dioxide ($CO_2$) and methane ($CH_4$) are significant players in the Earth's climate system, with 2016 being the first full year atmospheric $CO_2$ rose above 400 parts per million (ppm), with an average of $402.8 \pm 0.1$ ppm (Le Quéré et al., 2017). Since 1750 it has risen from 277 ppm. A similar trend has been seen with $CH_4$, increasing by 150 % in the atmosphere to 1803 ppb between 1750 - 2011 (Ciais et al., 2013), with an acceleration in recent years to 1850 ppb in 2017 (Nisbet et al., 2019). With the oceans being a sink for an estimated $\sim24\%$ of anthropogenic $CO_2$ emissions (Friedlingstein et al., 2019), they have been under continuous observation and study, resulting in the collection of large global databases (e.g., Takahashi et al., 2009;

Bakker et al., 2016). Such observations have shown both regional and/or temporal variabilities between a source and sink for $CO_2$, yet typically a low to moderate $CH_4$ source ($\sim$ 0.4-1.8 Tg $CH_4$ yr$^{-1}$; Bates et al., 1996; Borges et al., 2018; Rhee 2009), increasing in coastal regions (Bange, 2006). Inland waters however, are a different story and although it has been known for over 50 years that they are mostly supersaturated with $CO_2$ (Park, 1969), up until recently their budgets have been of relatively little focus. Although regions such as lakes, rivers and reservoirs, have been recognised as significant players in the carbon budget over the past couple of decades (see Carpenter et al., 1995, Cole and Caraco, 1998, Caraco, 2001 with updates and reviews from Tranvik et al., 2009, Raymond et al., 2013 and Regnier et al., 2013), compared to the ocean datasets, having quantified and consistent data and consensus within these regions is still in the relatively early stages. Global $CO_2$ and $CH_4$ emissions from inland waters are estimated at 2.1 Pg C yr$^{-1}$ (Raymond et al., 2013) and 0.7 Pg C yr$^{-1}$ (Bastviken et al., 2011) respectively. Mixing regimes (e.g. deltas and estuaries) as well as streams and smaller bodies of water are known to be overly important within these inland systems (Holgerson and Raymond, 2016; Natchimuthu et al., 2017; Grinham et al., 2018), yet there is very little data coverage with respect to both of these parameters (Borges et al., 2018), even more so when evaluated together. Therefore, a specific need exists for high-resolution spatiotemporal measurements in regimes of highly dynamic, varying $p$CO$_2$ concentrations (Yoon et al., 2016; Paulsen et al., 2018; Friedlingstein et al., 2019).

One issue leading to little data coverage is that the combination of both inland waters and the ocean, the Land-Ocean Aquatic Continuum (LOAC), are usually not studied continuously but rather split between oceanographers and limnologists. Although significant progress has been made recognizing the importance of the LOAC as a whole system (e.g. Raymond et al., 2013; Regnier et al., 2013; Downing 2014; Palmer et al. 2015; Xenopoulos et al., 2017), huge knowledge gaps are still present, particularly related to limited field data availability (Meinson et al., 2016). Often, this is due to different measuring techniques and protocols, both with respect to in situ/autonomous observations and the collection of discrete data. This has been previously noted, through 'blind' or 'spot' sampling having large effects on the overall measured results, potentially leading to under-/over-estimations in concentrations and fluxes (Richey et al., 2002; Abril et al., 2014; Canning et al., 2020). Furthermore, this is further complicated by $p$CO$_2$, $p$CH$_4$ and dissolved O$_2$ being controlled by several factors including biological effects, vertical and lateral mixing and temperature-dependent thermodynamic effects (Bai et al., 2015). These effects are exacerbated within inland waters where variability is far higher due to variations in environmental conditions and the magnitude of biological processes and anthropogenic influences (Cole et al., 2007). The high spatial and temporal variability within the inland/mixing waters (Wehrli, 2013) only increases these difficulties, ultimately leading to the interface between the ocean and inland to be considered one of the hardest systems to observe accurately and adequately. This has led to limitations and lack of verifications, leading to errors, discrepancies and uncertainties involved in scaling up the data. Inland waters tend to exhibit extreme ranges of $CO_2$ partial pressure ($p$CO$_2$, <100 to >10,000 $\mu$atm; this study; Abril et al., 2015; Ribas-Ribas et al., 2011) in comparison to oceanic waters ($\sim$ 100 – 700 $\mu$atm; Valsala and Maksyutov, 2010), while also showing extreme variabilities for both O$_2$ and $CH_4$. Given the much smaller concentration changes and gradients, oceanic sensors and methods have been specifically tailored to assure high accuracy over oceanic concentration ranges, in comparison to inland waters.

One way of tackling these limitations and measurement technique differences is through sensors, ensuring a unified way of measuring with well-constrained accuracy and precision. In specific regions, this has become more widespread and reviewed

numerous times within the coastal and open ocean (see Atamanchuk et al., 2015; Clarke et al., 2017). Multiple seagoing methods have been applied since the 1960's (see examples Takahashi 1961; DeGrandpre et al., 1995; Waugh et al., 2006; Pierrot et al., 2009; Schuster and Körtzinger, 2009; Becker et al., 2012) to measure and estimate greenhouse gases, such as $CO_2$ across a variety of aquatic regions. Inland water investigations have also seen clear progress with the development of

continuous, autonomous measurement techniques (e.g. DeGrandpre et al., 1995; Baehr and DeGrandpre, 2004; Crawford et al., 2014; Meinson et al., 2016; Brandt et al., 2017). Yet, only few studies have employed membrane-based equilibration sensors with NDIR detection (non-dispersive infrared spectrometry) (e.g. Johnson et al., 2009; Bodmer et al., 2016; Yoon et al., 2016; Hunt et al., 2017), with some adapting atmospheric sensors (see Bastviken et al., 2015). These methods often focus on only one gas (usually $CO_2$) and none of these methods mentioned, cover both water regimes (ocean to inland), potentially leading

to missed mixing regimes regions. On top of this, spatiotemporal data coverage has been noted to be sparse (Yoon et al., 2016) and is needed to advance our budget accuracies and understanding.

Given the biological and physical parameters of inland waters, multi-gas analyses is the way forward, which was previously noted by in the work of Brennwald et al., (2016), where they worked on the development of the membrane inlet mass-spectrometric (MIMS) 'miniRuedi'. This system measured as a nearly fully autonomous multi-gas mass spectrometer,

however despite advances in both inert and reactive gas measurements the need for a filter and gas standards for extreme gradients gives this set-up a disadvantage in highly diverse inland waters. This highlights one issue with extremely variable environments, and showing that there is need for developing a robust, fully autonomous sensor system that is portable enough for small and simple platforms. The development needs to be able to measure a full range of concentrations accurately and precisely which is usually out of the specifications of sensors designed for one region. It needs to have the potential to measure

multiple gases and ancillary parameters in unison across salinity and regional boundaries (including extreme concentrations), enabling us to measure throughout the LOAC. These efforts would hopefully bridge the ocean-limnic gap both technically and by reducing discrepancies and errors while having high-resolution, real-time measurements. To be accepted both within inland waters and the ocean, it needs to be within oceanic specifications while be able to handle larger concentration ranges. This is essential for improved monitoring, potentially avoiding 'spot' sampling bias, providing in-need high-spatiotemporal variability

data, tracking the global carbon budget (Le Quéré et al., 2017) and potential application in areas of highest uncertainties, with potentially high anthropogenic input (Schimel et al., 2016).

Here we used state-of-the-art membrane-based equilibrator NDIR (non-dispersive infrared) and TDLAS (tunable diode laser absorption spectroscopy) sensors for $pCO_2$ and $pCH_4$ respectively, an oxygen optode and a thermosalinograph to create a set-up allowing measurements in a continuous flow-through system. To assess the versatility, performance, portability and

measurement quality of the set-up, it was deployed across the three main aquatic environments: oceanic, brackish and limnic. We present the technical findings from the campaigns, showing the need for such high-resolution combined gas data on a larger spatiotemporal scale, however biogeochemical implications will not be further investigated. The primary objective of the work presented here was, with the use of oceanic, state-of-the-art tested sensors, realize a fully versatile, portable, and robust flow-through system to accurately and autonomously measure multiple dissolved gases ($CO_2$, $CH_4$, $O_2$), and ancillary parameters

(temperature, salinity) across the full range of salinities, simultaneously. The second was to assess the potential for high-quality

spatiotemporal data extraction. The set-up was subsequently deployed in each region of selected salinities (ocean, brackish and limnic waters) to allow for both spatial and temporal measurements. Extensive post field campaign corrections were assessed to see the need for their adaptation over all the regions, and for the more precise corrections, small-scale variability was used for this purpose. Discrete samples were collected, and reference systems were deployed alongside to provide quality assessment of the performance of the flow-through set-up.

## 2 Material and Methods

### 2.1 Sensors and ranges

The set-up featured four separate sensors measuring 3 dissolved gases as well as standard hydrographic parameters (Table 1).

The CONTROS HydroC® $CO_2$ FT (HC-$CO_2$) and CONTROS HydroC® $CH_4$ FT (HC-$CH_4$) (formerly Kongsberg Maritime Contros GmbH, Kiel, Germany; now -4H-JENA engineering GmbH, Jena, Germany; -4H-JENA) are both commercially manufactured sensors which use membrane-based equilibrators combined with NDIR and TDLAS gas detectors, respectively. Both sensors are of flow-through type, in which water is pumped through a plenum with a planar semi-permeable membrane across which dissolved gas partial pressure equilibrium is established with the head space behind, as described by Fietzek et al., (2014). The sensors were factory calibrated before and after each cruise (Romanian campaigns all together) and the calibration polynomials provided (in case of the $p$CO$_2$ sensor) by the manufacturer.

The CONTROS HydroFlash® $O_2$ (formerly Kongsberg Maritime Contros GmbH, Kiel, Germany; KMCON) was an optical sensor (optode) based on the principle of fluorescence quenching (see Bittig et al., (2018b) for an optode technology review). As the sensor was only available as submersible type, a flow-through cell was built around the sensor head for integration in the flow-through system.

The SBE 45 Micro Thermosalinograph (Sea-Bird Electronics, Bellevue, USA) was used to measure temperature and conductivity to calculate salinity. All sensor frequencies depended on cruise type and were set between 1 reading output per minute to 1 reading per second (oceanic to inland waters).

### 2.2 Initial procedures and background

Initial experiments were conducted within the laboratory at GEOMAR, Kiel, Germany and during short sea trials on-board RV Littorina in 2016 to ensure the optimal performance of all sensors (data not shown here). HC-$CO_2$ was placed within the set-up upstream of HC-$CH_4$ due to higher sensitivity and dependence of the parameter $p$CO$_2$ to temperature changes. The water flow was split between sensors due to differing flow range requirements. A flow meter and pressure valves were installed to provide optimal flow speeds, as shown in the schematic of the overall set-up (Fig. 1).

**Table 1.** Sensors and their manufacturer specifications, along with calibration range (for the CONTROS HydroC® $CO_2$ FT and $CH_4$ FT factory calibration ranges were specific for the campaigns).

| Model | Deployment Type | Detector Type | Resolution | Accuracy | Response Time ($t_{63}$) (min:sec) | Specified Flow Rate (L min$^{-1}$) | Power Consumption | Dimensions (mm) | Weight (kg) | Range of Factory Calibration |
|---|---|---|---|---|---|---|---|---|---|---|
| CONTROS HydroC® $CO_2$ FT (-4H-JENA)* | FT ** membrane equilibration | NDIR | <1 $\mu$atm | ±1 % of reading | $t_{63}$ ~ 1:32 @ 16°C, 5L min$^{-1}$ | 2 - 15 | 350 mA @ 12 V | 325 x 240 x 126 | 5.3 | 0 - 6,000 $\mu$atm |
| CONTROS HydroC® $CH_4$ FT (-4H-JENA)* | FT ** membrane equilibration | TDLAS | <0.01 $\mu$atm | ±2 $\mu$atm or 3 % of reading | $t_{63}$ ~ 22:46 @ 17°C, 7 L min$^{-1}$ | 6 - 15 | 600 mA @ 12 V | 452 x 283 x 142.5 | 8.5 | <2 - 40,000 $\mu$atm |
| CONTROS HydroFlash® $O_2$ (KMCON)*[1] | Submersible | Fluorescence Quenching | <0.1 % | ±1 % | $t_{63}$ < 00:03 | N/A | 0.1 J per sample | 23x197 with connector | 0.17 air 0.11 water | 0 - 400 mbar $pO_2$ |
| SBE 45 *** Thermosalinograph Conductivity | FT ** | Conductivity cell | 0.00001 S m$^{-1}$ | ±0.0003 S m$^{-1}$ | N/A | 0.6 - 1.8 | 30 mA @ 12-30 V | 338 x 134.4 x 76.2 | 4.6 | 0 to 7 S m$^{-1}$ |
| SBE 45 *** Thermosalinograph Temperature | FT ** | Thermistor | 0.0001 °C | ±0.002 °C | N/A | 0.6 - 1.8 | 30 mA @ 12-30 V | 338 x 134.4 x 76.2 | 4.6 | -5 to + 35 °C |

*-4H-JENA engineering GmbH, Jena, Germany; -4H-JENA (formerly Kongsberg Maritime Contros GmbH, Kiel, Germany)

*[1] formerly Kongsberg Maritime Contros GmbH, Kiel, Germany

** Flow-Through

*** Sea-Bird Scientific

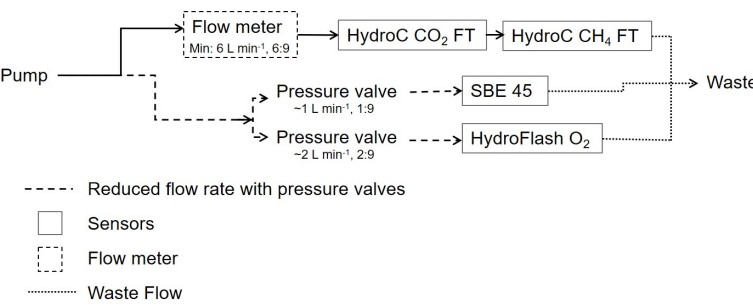

**Figure 1.** Flow schematic of the set-up, including minimal flow ratios. The flow rate was measured only for the main flow. For the side flows the rate was adjusted by pressure valves to be a fraction (i.e. 1:9 and 2:9 for L min$^{-1}$) of the total flow.

Depending on the vessel type and location of the measurement system therein, the pump was placed either in the 'moon pool' of the ship or at the front of the boat (limnic cruises, see Table 2). The total flow was regulated by multiple pressure valves to a pump rate of approximately 9 -10 L min$^{-1}$. The HC-CO$_2$ and HC-CH$_4$ show a distinct dependence of their response time (RT), on the water flow rate, with the demand for the HC-CO$_2$ flow rates ranging from 2-16 L min$^{-1}$ (manufacturer recommendation is 5 L min$^{-1}$) and for the HC-CH$_4$ flow rates from 6-16 L min$^{-1}$. Based on this information combined with preliminary testing and power accessibility considerations across all regions, 6 L min$^{-1}$ was used as the target flow rate for the HC-CO$_2$ and HC-CH$_4$. Data acquired with any flow rate below 5 L min$^{-1}$ were flagged questionable as may have contributed to increased errors.

The data was logged on an internal logger for the HC-CO$_2$ and HC-CH$_4$ in unison, and displayed live using the CONTROS Detect software. The SBE Thermosalinograph and HydroFlash O$_2$ were logged on SeatermV2 software and a terminal pro-gramme (Tera Term) respectively. The sensors have the capability to set the time-stamps for logged data, allowing alignment among all sensor systems and/or local time for discrete sample collection. Water flow was measured using LabJack Software and any power cuts (or other circumstances such as boats passing near to the house boat during the limnic cruises) were logged manually to ensure the best quality outcome from the data processing which is described in the next sections.

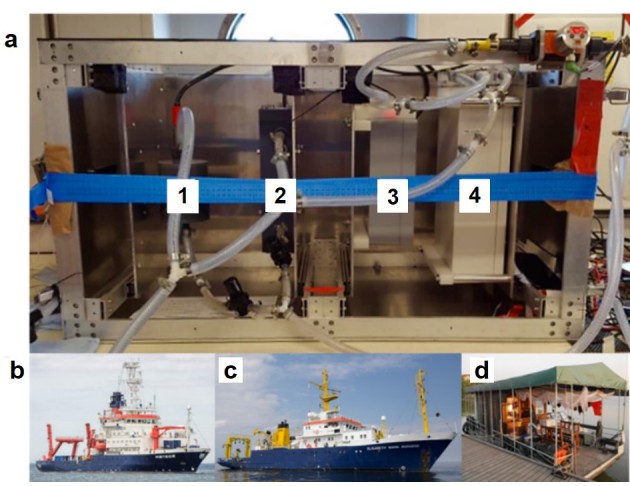

**Figure 2.** Complete flow-through set-up (a) in operation on board of RV Meteor indicating the easily accessible sensors for $O_2$ (1), T and S (2), $p$CO$_2$ (3) and CH$_4$ (4). Besides the operation on RV Meteor (b) across the Atlantic, M133, the set-up was also deployed on RV Elisabeth Mann Borgese (c) within the Baltic Sea, EMB 142, and on a houseboat in the Danube Delta (d), Romania, Rom1-3 for spring (Rom1), summer (Rom2) and autumn (Rom3).

The set-up was tested in three different locations: South Atlantic Ocean (oceanic), western Baltic Sea (brackish) and the Danube river delta (limnic), Romania between 2016 and 2017 (Fig. 2 and 3). Although, in brackish water regions like the Baltic Sea, the same measurement techniques for many instruments and sampling methods as within the ocean are used, certain techniques (e.g. alkalinity titration) generally need special adaption for the low-salinity range. The different deployments ensured the sensors were tested in the field across the full salinity range, from freshwater to seawater, from moderate to tropical temperatures and from low concentrations near atmospheric equilibrium to extreme cases of super- (CH$_4$, CO$_2$) or undersaturation (CO$_2$, O$_2$). The choice of cruises also allowed testing the versatility of the set-up with deploying it on a range of vessel types (Fig. 2 and Table 2 and section 2.3).

## 2.3 Campaigns

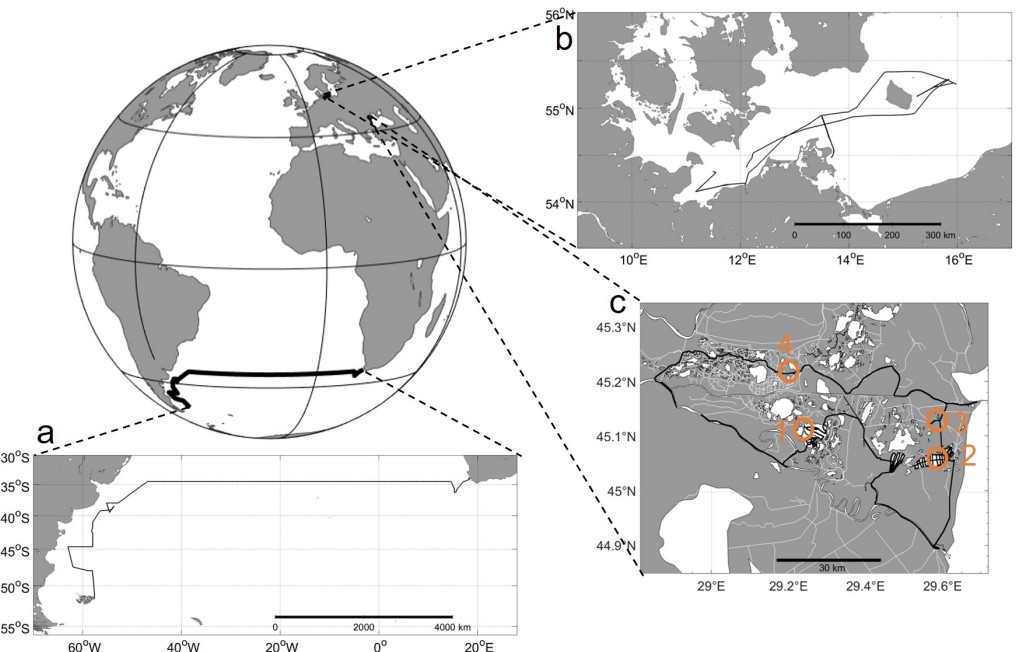

**Figure 3.** Transects for all test sites: (a) Oceanic: South Atlantic, RV Meteor cruise M133 Ocean (Cape Town, South Africa – Stanley, Falklands), (b) Brackish: RV Elisabeth Mann Borgese cruise EMB 142, Western Baltic Sea, (c) Limnic: Danube Delta, Romania, orange circles labelled 1 through 4, show stations of stationary overnight (c.f. Fig. 11). For further information see Table 2.

### 2.3.1 Meteor Cruise M133 to South Atlantic (oceanic)

The system was set up on the RV Meteor (cruise M133) during the SACROSS campaign, from Cape Town, South Africa to the Falkland Islands, UK between 15.12.2016 - 13.01.2017 (open to shelf oceanic waters). Discrete samples were collected throughout the cruise for total alkalinity (TA), dissolved inorganic carbon (DIC), $CH_4$ and $O_2$. The water was pumped up by means of a submersible pump installed in the ship's moon pool at about 5.7 m depth. The system logged once every minute which was deemed sufficient until the Patagonian Shelf was reached, where the measurement frequency was increased to 1 Hz. Sea surface temperature data was measured with a temperature sensor (SBE 38, Sea-Bird Electronics, Bellevue/WA, USA) installed at the seawater intake in the moon pool, which was used for temperature correction of the flow-through system data. Sea surface salinity was taken from the ship's thermosalinograph (SBE 21, SeaCAT TSG, Sea-Bird Scientific) located within the mess room, and the water inlet was located on the bulbous bow. This salinity data was used for the carbonate system calculations related to the discrete reference. $CH_4$ data collected during this cruise was not used due to an internal issue of the detector related to absorption peak identification. This data was automatically flagged within the sensor diagnostic values and

subsequently excluded. This issue was fixed for the limnic campaigns by installation of a reference gas cell in the absorption path of the detector.

### 2.3.2 Elisabeth Mann Borgese EMB 142 to Western Baltic Sea (brackish)

The sensor package was run on board of RV Elisabeth Mann Borgese (EMB 142) to the western Baltic Sea between 15.-22.10.2016 (brackish waters). The cruise was one of the main field activities of the Scientific Committee on Oceanic Research (SCOR) working group 142 (Dissolved $N_2O$ and $CH_4$ measurements: Working towards a global network of ocean time series measurements of $N_2O$ and $CH_4$) and entirely dedicated to the inter-comparison of continuous and discrete $N_2O$ and $CH_4$ measurement techniques (see Wilson et al., 2018), but some of the systems also measured $p$CO$_2$ continuously. Discrete samples were collected for validation of the $CO_2$ and $CH_4$ sensors. All analyzers and the discrete sampling line were connected to the same water supply from a submersible pump system installed in the ship's 'moon pool' (depth 3 m), ensuring that the same water was used by all groups. A back-pressure regulation system assured independent flow assurance of the individual set ups. The sensors logged continuously at a rate of between once per second and once per minute, depending on local variability. During this cruise, only half the $CH_4$ data was used due the same technical reason as stated for the oceanic cruise.

### 2.3.3 Romania 1-3 Cruises to the Danube River Delta, Romania (limnic)

Campaigns over three consecutive seasons were conducted during three field campaigns throughout the Danube Delta in Romania (limnic) in 2017: during spring (Rom1: 17.-26.06.2017), summer (Rom2: 03.-12.08.2017) and fall (Rom3: 13.-23.10.2017). The Danube Delta is situated at the border of Romania and Ukraine on the edge of the Black Sea. It is the second largest river delta in Europe with a diverse wetland area of about 3,000 km$^2$ with a variety of lakes, rivers and channels. The equipment was set-up on board a small houseboat, giving access to smaller channels and 'hard to reach' areas. A small power generator or car batteries were used to power the system. With an 11-24 V power source the set-up can take a reading at up to 1 Hz. In combination with the flow-through set-up, discrete samples were collected using the same water inlet as the sensors prior to the sensor inlet with little disruption to the overall water flow. Data acquisition was only interrupted when there were unexpected rainstorms or problems with the power supply, i.e. power cuts due to lack of fuel. Bilge pumps were deployed from the bow of the house boat to reduce water body perturbations caused by the boat that would affect the flow-through measurements. The excess water was discarded over the side, away from the pump location. A few times during the deployment, there was no SBE data since its data logging did not automatically restart after a re-powering of the system. During these times, temperature data from the optode (mean offset from the SBE for all cruises $0.16 \pm 0.10$ °C) was used instead.

### 2.4 Method Validation

To validate the sensor measurements, discrete samples were collected simultaneously from the same water source (vessel-dependent) using tubing connected to the manifold to which the sensors were connected to.

TA and DIC samples were collected in 500 mL Duran glass bottles (100 mL borosilicate glass bottles for inland waters) following the standard operating procedure for water sampling for the parameters of the oceanic carbon dioxide system (SOP 1, Dickson et al., 2007) with 87, 8 and 68 discrete samples from the oceanic, brackish and inland water cruises, respectively. The samples were poisoned with 100 $\mu$L (20 $\mu$L inland) of saturated $HgCl_2$ solution to stop biological activity from altering the carbon distributions in the sample container before analysis, a procedure not typically performed in limnic research. A headspace of approximately 1% of the bottle volume was left to allow for water expansion. A greased stopper was put in place and secured in an airtight manner using an elastic strap. The samples were then stored in a dark, cool place until measured. The VINDTA (Versatile Instrument for the Determination of Titration Alkalinity, Marianda Analytics and Data, Kiel, Germany) and SOMMA (Single Operator Multiparameter Metabolic Analyzer, University of Rhode Island, Narragansett Bay/MA, USA) were used to measure TA (Mintrop et al., 2000) and DIC (Johnson, 1987) in the brackish and seawater samples. Freshwater samples were measured using the Apollo Total Alkalinity Titrator (Model AS-ALK2, Apollo SciTech, Newark, USA) and DIC Analyzer (Model AS-C3, Apollo SciTech, Newark, USA). Measurements were calibrated with certified reference material (CRM) provided by A. Dickson (University of California, San Diego/CA, USA) with a determined precision of $\pm1.64$ $\mu$mol $kg^{-1}$ and $\pm1.15$ $\mu$mol $kg^{-1}$, respectively for DIC and TA and freshwater precision from duplicates were $\pm1.29\mu$mol $kg^{-1}$ and $\pm2.90\mu$mol $kg^{-1}$ for DIC and TA respectively.

TA and DIC were then used to compute $p$CO$_2$, using the open-access software CO2SYS software (Lewis et al., 1998) employing the Millero (2010), Millero et al., (2006) and Millero, (1979) carbonic acid dissociation constants (K1 and K2) for seawater, brackish and freshwater samples, respectively. For the pH scale and $KSO_4$ dissociation constants, seawater and Dickson and Riley, (1979), were used.

CH$_4$ samples were collected in 20 mL bottles, poisoned with 50 $\mu$L of saturated solution $HgCl_2$ and crimp-sealed. The samples were then stored until measurement. CH$_4$ in these water samples was measured with a gas chromatographic method following a procedure described by Weiss and Price (1980) and A. Kock (unpubl.) with an average standard deviation of the mean CH$_4$ concentration of 2.7% calculated following A. Kock (unpubl.) and David (1951). During transportation and storage, some CH$_4$ samples developed air bubbles due to warming causing some of the gases (e.g., nitrogen, oxygen) to become supersaturated and eventually out-gas; these samples were discarded.

During the brackish water cruise, the mobile equilibrator sensor system (MESS: Leibniz Institute for Baltic Sea Research) was used as a reference system. The system consists of an open mixed showerhead-bubble type equilibrator, with an auxiliary equilibrator attached to the main exchange vessel. Water flow was adjusted to approximately 6 L min$^{-1}$ during the cruise. Three Cavity Enhanced Absorption Spectrometers (CEAS) were attached in parallel from which only the results of the Los Gatos Research (LGR) GHG analyzer (Los Gatos Research, San Jose, California, USA) determining $x$CO$_2$ and xCH$_4$ are used for the comparison purposes in this study. Total air flow through the pumps of the sensors and an additional air pump was set to approximately 1 L min$^{-1}$. A set of calibration gas runs covered a range from 1,806 to 24,944 ppb for methane and 201.3 to 1,001.5 ppm for CO$_2$. Source of the calibration gases was the central calibration facility of the European Integrated Carbon Observation Research Infrastructure (ICOS CAL). The high standard was produced by NOAA as initiative of the SCOR working group 143. The response time for methane and CO$_2$ for the chosen flow rates were determined prior to the

cruise to be approximately $\sim 330$ s and $\sim 35$ s respectively, at roughly 6 L min$^{-1}$ with a gas flow of 4.7 L min$^{-1}$. Similar system operations and details of the post processing of data are given in Gülzow et al., (2011), which is installed on a VOS line and regularly reporting the data to the SOCAT data base (Bakker et al., 2016).

Oxygen was sampled in 100 mL borosilicate glass bottles with precisely known volume and titrated using the Winkler Method (Winkler, 1888) on the oceanic cruise. The precision of the Winkler-titrated oxygen measurements was 0.29 $\mu$mol L$^{-1}$ and based on 120 duplicates, from the mathematical average of standard deviations per replicate. Samples containing any air bubbles were discarded.

## 2.5 Sensor data processing

The corrections on the raw $p$CO$_2$ output from the HC-CO$_2$ sensor were for sensor drift (section 2.5.1: both zero and span), any observed warming of the sampled water at the sensor with respect to the seawater intake temperature (section 2.5.2), extended calibrations (section 2.5.3: over 6,000 ppm, i.e. upper limit of manufacturer calibration range, although calibrated with $x$CO$_2$, final data by the sensors was converted to $p$CO$_2$) and the effect of the sensor response time (RT)(section 2.5.4).

### 2.5.1 Sensor drift

Sensor drift for the HC-CO$_2$ was corrected on the basis of pre- and post-deployment calibrations and the regular in situ zeroings using the sensor's auto-zero function, in which CO$_2$ is scrubbed from the measured gas stream using a soda lime cartridge. This zero measurement is then used in post-processing to correct for the drift over the deployment, details of which are described in Fietzek et al., (2014). The zeroings were carried out at regular intervals of 4 to 12 h in the various field campaigns and for correction during processing, we considered temporal change in the concentration-dependent response of the sensor between pre- and the post-cruise factory calibration, i.e. span drift, to be linear to the sensor's runtime.

### 2.5.2 Temperature correction

The temperature correction was applied for all $p$CO$_2$ data to correct for any temperature difference between measurement in the flow-through setup and in situ temperature. After a time-lag correction due to in situ temperature and equilibrium temperature mismatch, resulting from the travelling time of the water from intake to sensor spot, the (Takahashi et al., 1993) temperature correction was used for $p$CO$_2$:

$$p\text{CO}_2(\text{T}_{\text{insitu}}) = p\text{CO}_{2(\text{T}_{\text{equ}})} \cdot \exp[0.0423 \cdot (\text{T}_{\text{insitu}} - \text{T}_{\text{equ}})] \tag{1}$$

where T$_{insitu}$ is the in situ temperature (i.e. sea surface temperature from inlet: SST), and T$_{equ}$ is the equilibration temperature. For CH$_4$, the correction following Gülzow et al., (2011) was applied:

$$p\text{CH}_4 = p\text{CH}_{4,\text{equ}} \cdot \left( \frac{\text{CH}_{4,\text{sol,equ}}}{\text{CH}_{4,\text{sol,insitu}}} \right) \tag{2}$$

where $pCH_4$ is the final $pCH_4$ (atm), $pCH_{4,equ}$ is the $pCH_4$ (atm) at equilibrium, $CH_{4,sol,equ}$ is the solubility (mol (L $\cdot$ atm)$^{-1}$) of $CH_4$ at equilibrium temperature and $CH_{4,sol,insitu}$ is the solubility (mol (L $\cdot$ atm)$^{-1}$) at in situ temperature.

### 2.5.3 Extended calibrations

During the Danube river field campaigns, $CO_2$ data sometimes exceeded 6,000 ppm, i.e. the upper limit of the factory calibration range. NDIR detectors such as the one used in the HC-$CO_2$ sensor, show a non-linear signal response, therefore extrapolation over the factory-calibrated range could not be done safely and an extended calibration was conducted. Prior to the extended calibrations, a further 'post' processing calibration was conducted by the manufacturer. The polynomial was compared to that of the initial post calibration from Rom2, revealing an average offset between the two of - 0.766 $\pm$ 0.94 ($\pm$ SD) ppm. This proved that the HC-$CO_2$ had shown little change over the period, ensuring the extended calibration was still applicable and could be applied to this campaign. The extended lab calibration was performed on manually produced gas mixtures. The $xCO_2$ of these mixtures was calculated considering the precisely measured flow ratios of the mixed gases $N_2$ and $CO_2$. The prepared calibration gas was wetted and routed to the HC-$CO_2$ membrane equilibrator. An extended calibration curve was then estimated to reduce the measurement uncertainty over an extended range 5,000-30,000 ppmv. Still, the measurement error at this range of approximately 3 % is larger then the $\pm 1\%$ accuracy for measurements within the regular factory calibration range. This was due to larger uncertainties of the calibration reference ($N_2$ and $CO_2$: AIR LIQUIDE Deutschland GmbH, Düsseldorf, Germany, 99.999 % and 99.995 % accuracy respectively), the flow error of the mass flow controllers and the smaller sensor sensitivities at higher partial pressures. Although calibrated with $xCO_2$, final data by the sensors was converted to $pCO_2$ in $\mu$atm.

### 2.5.4 Response time

The HC-$CO_2$ sensor response time (RT) for the corresponding flow rate and temperature was estimated from the signal recovery after each zeroing interval, by fitting an exponential function to the signal increase following the zeroing. Sensor response time is typically denoted as $t_{63}$, which represents the e-folding time scale of the sensor, i.e. the time over which, following a stepwise change in the measured property, the sensor signal has accommodated 63 % of the step's amplitude (Miloshevich et al., 2004). This correction was carried out by following a RT correction (RT-Corr) routine by Fiedler et al., (2013). However, the conditions within the limnic regions were simply too variable compared to the available in situ RT determinations and the in situ RT-dependencies that could be derived from the in situ measurements. Therefore, prior to the first campaigns, experiments were conducted within temperature-controlled culture rooms to see how the e-folding time of the HC-$CO_2$ flow-through sensor was affected by flow and temperature. These characterizations were used as the basis for the HC-$CO_2$ RT-Corr for the limnic cruises, as described below for HC-$CH_4$. Procedures for RT-Corr are further described by Fiedler et al., (2013) and Miloshevich et al., (2004).

Due to the HC-$CH_4$ using a TDLAS detector, drift correction was not needed as it produces a derivative signal that is directly proportional to $CH_4$ eliminating offsets in a 'zero baseline technique', along with a narrow band detection, therefore reducing signal noise (Werle, 2004). However, compared to the HC-$CO_2$, the HC-$CH_4$ sensor has an approx. 15 times longer RT due

multiple combined reasons: lower solubility, lower $CH_4$ permeability of the membrane material, and the comparatively larger internal gas volume. To enable a meaningful analysis of all the dissolved gas sensor signals, the $CH_4$ data therefore needed a RT-Corr. This was derived in a different way to the HC-$CO_2$, as no zeroing process within the HC-$CH_4$ allowed regular phenomenological estimation of the in situ RT during measuring. To quantify the $CH_4$ sensor's $t_{63}$, laboratory experiments with a modified sensor unit were conducted at different flow rates (5.7, 6.5 and 7 L min$^{-1}$) and temperatures (11.06, 15.05,

18.04 ° C) to determine the RT as a function of these parameters. The HC-$CH_4$ used for the RT determination experiments was modified by the installation of two additional valves in the internal gas circuit (c.f. the sensor schematic in the Fietzek et al., (2014)). Switching these valves enables bypassing the membrane equilibrator and causes e.g. equilibrated, low $pCH_4$ gas to be continuously circulated through the detector. Then the $pCH_4$ in the calibration tank could be increased. As soon as a stable $pCH_4$ level was reached in the tank, the valves within the HydroC were switched back and the gas passes the membrane

equilibrator again. From the resulting signal increase the time constant for the equilibration process i.e. the sensor RT could be determined.

    These modifications only affected the internal gas volume and flow properties to a small extent. An effect on the determined RT compared to the RT of a standard HC-$CH_4$ is therefore considered negligible. This information was then applied to the raw HC-$CH_4$ field data considering the measured flow rate and temperature and the method of (Fiedler et al., 2013).

Post processing of the HC-$O_2$ followed the SOP provided by KMCON using Garcia and Gordon (1992) combined fit constants. Further processing to convert the output into gravimetric ($\mu$mol kg$^{-1}$) and volumetric units ($\mu$mol L$^{-1}$) for comparison with other sensors and the discrete samples is described in the SCOR WG142 recommendations on $O_2$ quantity conversions (Bittig et al., 2018a). During the oceanic cruise, an average offset of -19.04 $\pm$ 2.26 ($\pm$ RMSE) $\mu$mol L$^{-1}$ and -29.78 $\pm$ 5.04 ($\pm$ RMSE) $\mu$mol L$^{-1}$ was also found within the open ocean and shelf respectively to discrete oxygen data. Two separate linear

offset corrections were applied throughout all oceanic data. Significant optode sensor drift, particularly when the sensor is not in the water, is a well-documented phenomenon (Bittig et al., 2018b).

    The output from the factory-calibrated SBE 45 thermosalinograph had no need for post processing. All accuracies of the sensors are shown in Table 1.

## 3   Results

The set-up was easily adapted to each power source, managing to measure across the range of salinities and concentrations (Table 2). The results of the corrections are shown in the following sections.

### 3.1   Extended calibration

Compared with the prior calibration curve from the conducted calibration at KMCON, the final extended 5-degree polynomial

had to be shifted slightly (690 ppm). This was expected due to slightly different calibration methods, so that both polynomials matched at the top of the calibration range from KMCON ($\sim$ 6,000 ppm). The sensor was able to reach values of nearly 30,000

**Table 2.** Cruise table for all field campaigns in 2016 and 2017, with cruise/ship names (cruise ID in bold), areas, as well as observed maximum to minimum values for all measured parameters (in bold). For $p$CO$_2$, the sensor is only factory-calibrated up to 6,000 ppm; therefore this was deemed as the maximum in these circumstances.

| | Cruise Information | | | | Observed Parameter Ranges | | | | |
| --- | --- | --- | --- | --- | --- | --- | --- | --- | --- |
| Cruise ID | Location | Vessel Size (m) | Campaign Date (d.month.yr) | | $p$CO$_2$ ($\mu$atm) min – max | $p$CH$_4$ ($\mu$atm) min – max | O$_2$ ($\mu$mol/kg) min – max | Temp (°C) min – max | Salinity min – max |
| RV Elisabeth Mann Borgese (**EMB 142**) | Baltic Sea, brackish | | 15.10.2016 to 22.10.2016 | | 378 – 576 | **2** – 7 | 268 - 304 | 11.0 – 13.4 | 7.40 – 15.9 |
| RV Meteor (**M133**) | South Atlantic Ocean | 98 | 15.12.2016 to 13.01.2017 | | 215 – 429 | N/A | 218 – 306 | **8.5** – 23.3 | 33.3 – **36.3** |
| Romania (**Rom1**) Spring | Danube Delta, limnic | ~10 | 17.05.2017 to 26.05.2017 | | **14** – > **6,000** | 76 – 8,660 | 173 – **431** | 14.2 – 23.2 | 0.16 – 0.25 |
| Romania (**Rom2**) Summer | Danube Delta, limnic | ~10 | 03.08.2017 to 12.08.2017 | | 25 – > **6,000** | 118 – **11,700** | 27 – 378 | 25.4 – **33.4** | **0.16** – 0.37 |
| Romania (**Rom2**) Fall | Danube Delta, limnic | ~10 | 13.10.2017 to 21.10.2017 | | 178 – > **6,000** | 104 – 9,430 | **7** – 377 | 14.2 – 17.8 | 0.18 – 0.26 |

ppm before starting to reach saturation (Fig. 4). Given this range was of similar magnitude as discrete samples and previous $p$CO$_2$ data from the Danube Delta (M. S. Maier unpubl.: reaching values of up to and over 20,000 ppm), the correction was applied to all data above 6,000 ppm. However, note the general uncertainty of the sensor at this range is larger than that at company 'operational values' and due to the longer time period between deployments, spring and summer campaigns have an unquantified increased error. We assume 3 % as a conservative estimate of the overall accuracy of the $x$CO$_2$ measurements in the extended range (> 6,000 ppm). From the noise of the signal during the calibrations, the estimated precision is $\pm$ 1 % of the CO$_2$ reading and we think even at this reduced accuracy the observations in the high $p$CO$_2$ range are of significant scientific value.

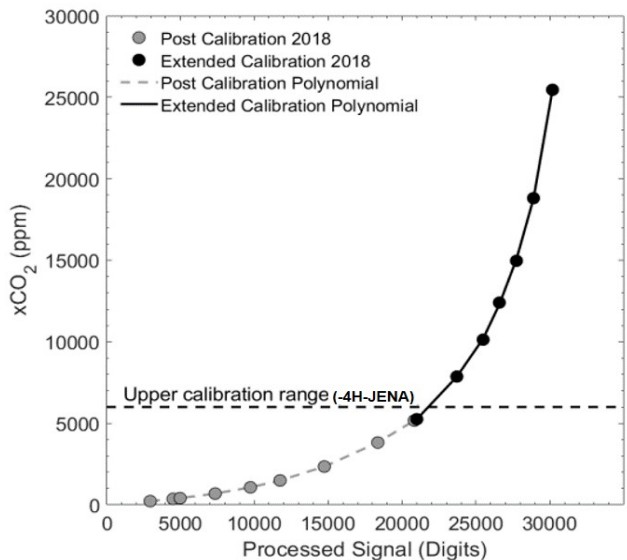

**Figure 4.** Calibration polynomial from the post-cruise manufacturer calibration (grey dots) (KMCON) and the manual extended calibration curve (black dots), with the top range of KMCON calibration range indicated (dashed line) above which the non-linear behavior of the NDIR (non-dispersive infrared spectrometry) sensor becomes stronger. Processed signal was calculated from the raw and reference signal data during processing.

## 3.2 RT-Correction analysis

The HC-CO$_2$ responce time correction (RT-Corr) laboratory experiments quantitatively show the effect of temperature and flow and point to the importance of recording the flow data (Fig. 5). As stated before, due to varying flow and temperature, the HC-CO$_2$ RT was determined by the laboratory experiments shown in Fig. 5a. An example of the estimation of $t_{63}$ is given in Fig. 5b, which shows the signal recovery following a 'zeroing' procedure, with $t_{63}$ = 93 s. Both, increased flow rate and temperature reduce the RT of the sensor significantly.

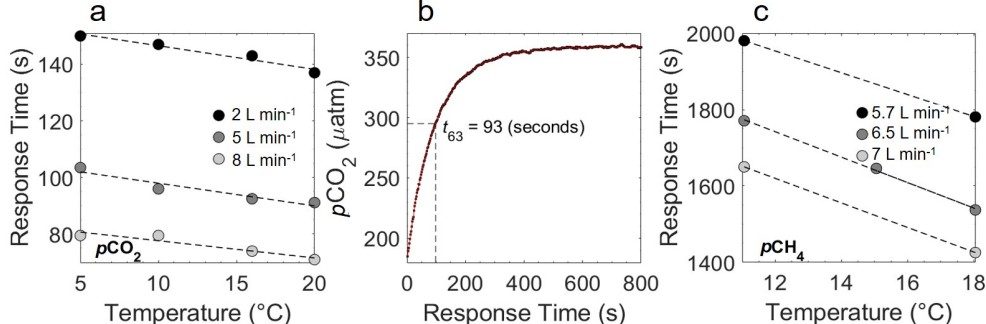

**Figure 5.** Response Time (RT) (s) of the HC-CO$_2$ determined at 4 temperatures within controlled laboratory conditions for different water flow rates (2, 5 and 8 L min$^{-1}$) with errors too small to see (a). An example of the output (b) shows how $t_{63}$ is retrieved after a zeroing interval with $t_{63}$ determined by the models fit (fit line in red). RT for $p$CH$_4$ also shown (c) for flow rates of 5.7, 6.5, and 7 L min$^{-1}$ conducted in controlled conditions at KMCON.

The RT of the HC-CH$_4$ was far higher than for CO$_2$ and varied between 1,425 – 1,980 s (Fig. 5c) depending on temperature and flow, with both higher temperature and flow rate yielding shorter RT (for comparison, $t_{90}$ for the HC-CO$_2$ and HC-CH$_4$ was 212 and 3,145 seconds respectively). This was then applied to the raw HC-CH$_4$ data and compared with the $p$O$_2$, which has a RT of < 3 s (KMCON, HydroFlash user manual) and therefore does not require an RT correction, to qualitatively assess the suitability of the correction, which can be seen from the near-perfect qualitative match between $p$CH$_4$ and $p$O$_2$ (Fig. 6). Note the inverted $p$O$_2$ due to typically having an inverse relationship with $p$CH$_4$.

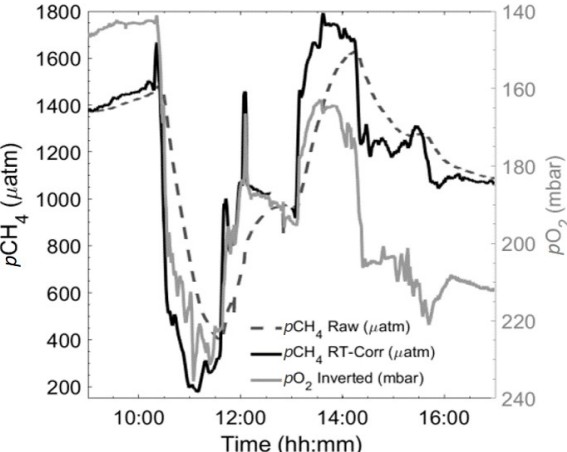

**Figure 6.** Section of a 24 h-cycle of data from the autumn limnic cruise (Rom3), showing raw (black dashed) and RT-Corr (black solid line) $p$CH$_4$ $\mu$atm measured by the HC-CH$_4$ with inverted $p$O$_2$ mbar (grey) as a technically independent yet parameter-wise linked reference for 'real-time' spatiotemporal variability, i.e. RT O$_2$ sensor « RT CH$_4$ sensor.

### 3.3 Verification by discrete sample comparison

#### 3.3.1 CO$_2$

Discrete $p$CO$_2$ was calculated from TA and DIC measurements, that had an average precision from replicates of 1.48 $\mu$mol kg$^{-1}$ (TA) and 1.04 $\mu$mol kg$^{-1}$ (DIC) after removal of one outlier sample. During the oceanic cruise, this provided a mean difference within the open ocean of -0.13 $\pm$ 5.25 $\mu$atm ($\pm$SD) to the data measured by the HC-CO$_2$ flow through system (HC-CO$_2$ $p$CO$_2$ – calculated $p$CO$_2$ [DIC and TA]). This mean increased within the productive waters along the Patagonia Shelf waters to up to 2.56 $\pm$ 6.21 $\mu$atm ($\pm$SD). A comparison between the performances of the HC-CO$_2$ is shown in Fig. 7, where each region has been separated. The comparison for the brackish water is against the calibrated data from the state-of-the-art equilibrator set up using an LGR oa-ICOS (Gülzow et al., 2011) showing an offset of -2.87 $\pm$ 7.71 ($\pm$SD) $\mu$atm (HC-CO$_2$ – reference $p$CO$_2$). Note the change in $p$CO$_2$ for each region, varying from under saturated (mainly oceanic waters) to supersaturated (brackish waters) and almost 20,000 $\mu$atm within the limnic waters during Rom1 (Fig. 7).

The limnic measurements of TA and DIC themselves had an average precision based on replicates of 1.03 $\mu$mol kg$^{-1}$ and 0.27 $\mu$mol kg$^{-1}$ for TA and DIC respectively.

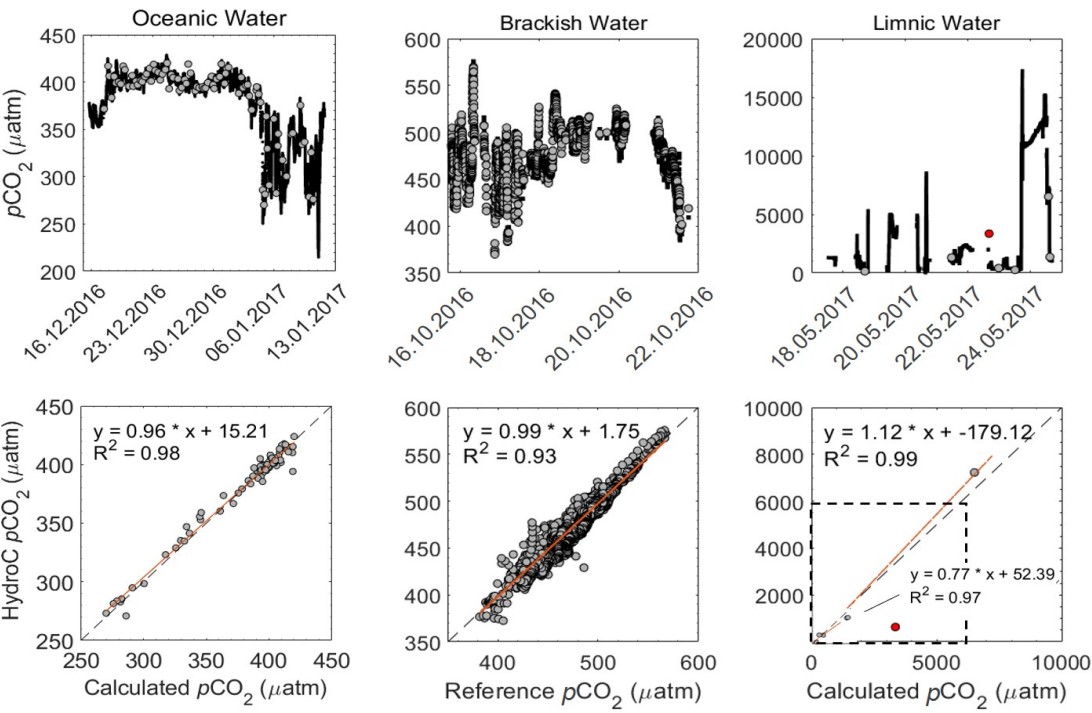

**Figure 7.** $pCO_2$ ($\mu$atm) from the HC-$CO_2$ within the 3 regions: oceanic (a and d: including the open ocean and the Patagonian shelf), brackish (b and c) and limnic waters (c and f) with the reference data used. The top graphs show the overall transects with HC-$CO_2$ data as black line and the reference data as grey dots (date as d.month.yr). The lower are property-property plots showing the 1:1 line (dashed) and line of best linear fit (orange). For validation of our system we used calculated $pCO_2$ from TA and DIC using $CO_2$SYS for both oceanic and limnic waters, whereas a reference system for the brackish waters as described above. During the Rom1 (limnic cruise), n = 7 with the outlier in red (sample with unclear match to flow-through data excluded from fit), with the box (dashed) indicating the 6,000 ppm company calibration limit.

### 3.3.2 CH$_4$

The average offset of the reference system to the HC-CH$_4$ during the first half of the brackish cruise was -0.95 ± 0.19 (±SD) $\mu$atm for the RT-Corr $pCH_4$ data (Fig. 8). This gave an average offset within the manufacturer accuracy specification range of ±2 $\mu$atm, with a mean offset of 0.79 ± 0.64 (±SD) $\mu$atm. Both sensors showed the same variability and magnitude to one another even with the offset (Fig. 8).

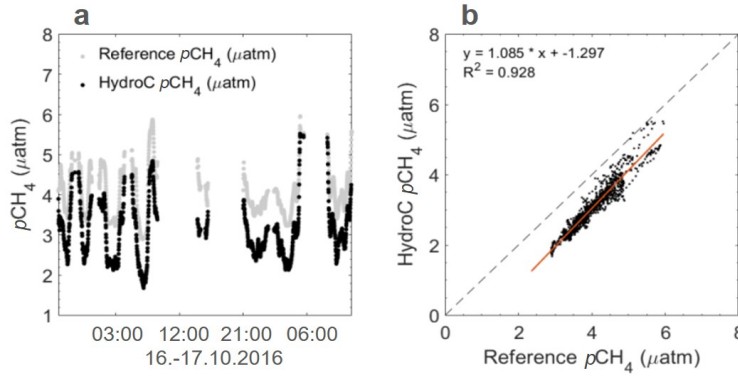

**Figure 8.** $p$CH$_4$ $\mu$atm data from the HC-CH$_4$ during the brackish cruise (EMB 142: R/V Elisabeth Mann Borgese) expedition in the Western Baltic Sea with the reference system over the first half of the cruise. a) shows the HC-CH$_4$ data with a negative offset resulting in lower concentrations compared to that of the reference data. b) gives a 1:1 plot with regression line (R$^2$ = 0.928), illustrating the constant offset, but similar slope of data. Within the brackish waters, the offset was within the specifications from KMCON, yet both values (reference and HC-CH$_4$) were in the range of that previously found within the region (Gülzow et al., 2013).

Given the previous evaluation of the RT-Corr, this improved the accuracy of the HC-CH$_4$ within the limnic system of Romania (HC-CH$_4$ – measured $p$CH$_4$: Rom1: from -164.3 $\pm$ 1,117.3 ($\pm$SD) $\mu$atm to 182.6 $\pm$ 591.3 ($\pm$SD) $\mu$atm; Rom2: from 609.3 $\pm$ 1,065 ($\pm$SD) $\mu$atm to 537.9 $\pm$ 1,145 ($\pm$SD) $\mu$atm and Rom3: from 466.5 $\pm$ 383 ($\pm$SD) $\mu$atm to 457.1 $\pm$ 376 ($\pm$SD) $\mu$atm (Rom1 shown in Fig. 9). Matching discrete sample data with continuous sensor data that has a long RT, becomes very complicated in highly variable situations. The effect of variable situations was also noticeable within the triplicates of
the discrete samples, some varying by over 400 $\mu$atm (with an average variability between repeated samples at 122.6 $\pm$ 100.9 ($\pm$SD) $\mu$atm), leading to the offset with the HC-CH$_4$ seeming reasonable. The agreement between sensor data and discrete samples increased significantly with the RT-Corr, shown in Fig. 9 with the R$^2$ improving from 0.33 to 0.93 and the slope from 0.36 to 1.25. Peaks within the data are also observed within the discrete measurements (Fig. 9 bottom) in combination with the sensor data e.g. in the lower graph of Fig. 9 between 20.05 and 21.05. It has to be noted that the determination of dissolved
CH$_4$ concentrations from discrete samples is also not fully mature and shows significant inter-laboratory offsets (Wilson et al., 2018) and thus, the observed discrepancy is likely not to be entirely caused by our sensor-based measurements.

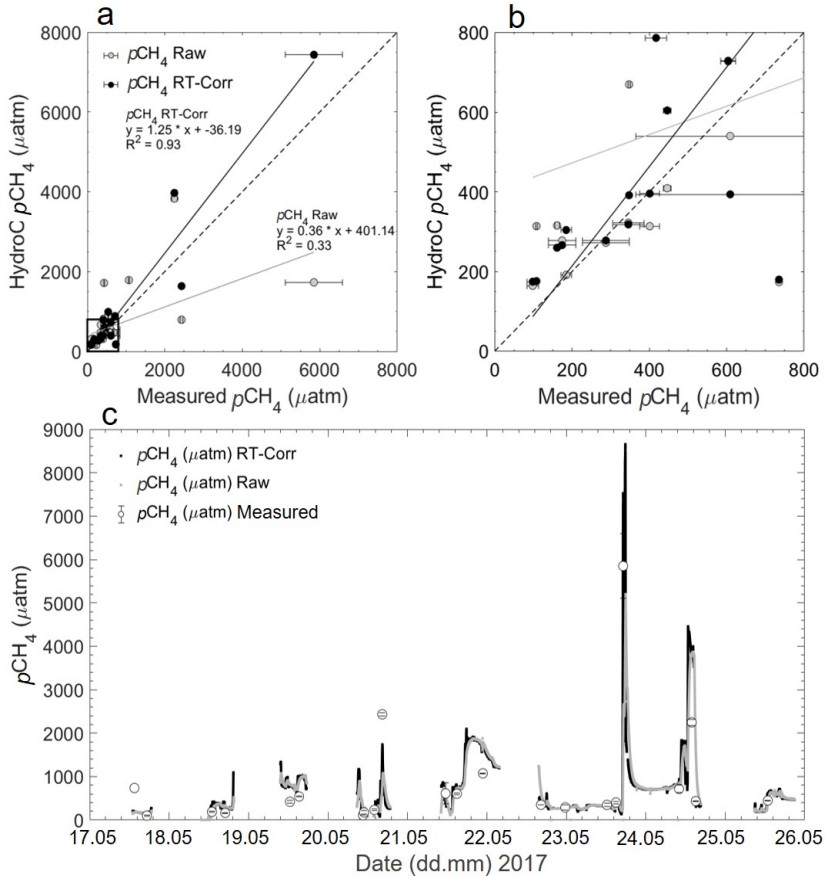

**Figure 9.** (a) Rom1 $pCH_4$ $\mu$atm data versus measured discrete samples of $pCH_4$ $\mu$atm with both raw HC-CH$_4$ data and response time corrected (RT-Corr) data over the full range of concentrations. (b) A close up view of the lower 800 $\mu$atm with errors for the measured samples against the HC-CH$_4$ data. The grey line signaling the line of best fit for the raw $pCH_4$ and black line signaling the RT-Corr $pCH_4$ $\mu$atm. (c) Full transect with discrete $pCH_4$ $\mu$atm samples for the spring cruise over time (date in dd.mm), some error bars too small to see.

### 3.3.3   O$_2$

During the oceanic cruise, after the post offset-correction (stated above), O$_2$ $\mu$mol L$^{-1}$ had an average offset of -0.1 $\pm$ 3.4 ($\pm$SD) $\mu$mol L$^{-1}$ (HydroFlash O$_2$ – discrete samples O$_2$) over the whole transect (Fig. 10). Altough stable and matching the

variability throughout (Fig. 10a), note the increased offset observed when entering the Patagonian Shelf (Fig. 10b).

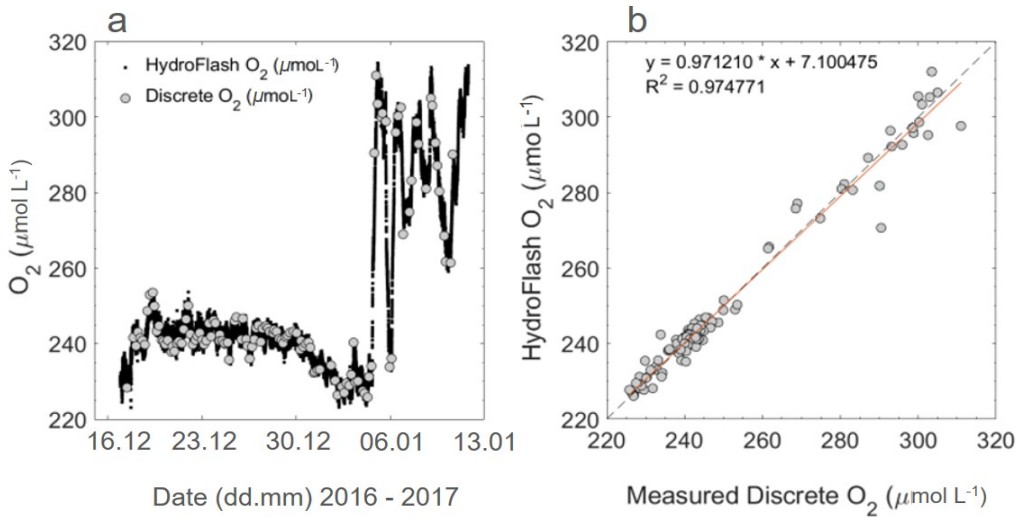

**Figure 10.** Oxygen concentration during M133 from re-calibrated continuous optode and discrete Winkler titration measurements (a), and property-property plot (b). For the open ocean and shelf waters the mean offset is -0.08 $\pm$ 1.89 ($\pm$SD) $\mu$mol L$^{-1}$ and -0.15 $\pm$ 6.49 ($\pm$SD) $\mu$mol L$^{-1}$, respectively. Higher variability towards the end of the transect is due to entering the productive Patagonian Shelf (d.month.yr: 06.01.2017 - 13.01.2017), also seen in higher concentrations of $O_2$.

## 4  Discussion

We have presented a portable, easily accessible, quick to set up multi-gas measurement system that can autonomously measure across the entire LOAC. The operational boundaries of these sensors were tested over various deployment durations ($\sim$ 1 month to hours), small spatial scales and under a wide range of operational environmental conditions.

Oceanic $p$CO$_2$ sensors are needed to operate with an overall accuracy of $\pm$ 2 $\mu$atm (Pierrot et al., 2009), therefore this sensor performance throughout the open ocean was considered very good (section 3.3.1). The offset found during the oceanic campaign when entering the Patagonian shelf (5.26 $\pm$ 4.33 $\mu$atm), is potentially due to biofouling within the tubing from the pump to the sensors. The offset observed by the optode for $O_2$, increased during the Patagonian Shelf waters due to the higher concentration ranges and gradients found along the shelf, possibly indicating an emerging biofouling issue of the sensor or

within the casing surrounding the sensor. This demonstrated the overall relatively long-term stability and reliability of the $O_2$ optode even in an area with such extreme hydrographic variability. This was expected due to optodes being used widely in multiple environments (see Bittig et al., (2018b); Kokic et al., (2016); Wikner et al., (2013) for oceanic, coastal and fresh water examples).

     In the brackish water campaign, the HC-CO$_2$ and HC-CH$_4$ showed good agreement with the reference systems, within the

380 manufacturers specifications. The data from the HC-CH$_4$, although having an internal issue as stated in section 2.3.1, showed

the same magnitude and variability as the reference system (Fig. 8). With little noise from both systems, natural varability was witnessed by both to further assure the system was running efficiently.

The limnic campaigns were ideally suited to test the flexibility of these sensors, with concentration ranges reaching almost 30,000 $\mu$atm for $p$CO$_2$, over 10,000 $\mu$atm for $p$CH$_4$, and O$_2$ ranging from supersaturated to suboxic. Direct comparisons with the CH$_4$ and CO$_2$ concentrations show relatively similar variations as previous measurements within the Danube Delta lakes (Durisch-Kaiser et al., 2008; Pavel et al., 2009). Due to the design, physical placement and high flow speed, no biofouling of the membranes of the HC-CO$_2$ and HC-CH$_4$ occurred even within particle-rich environments, with very little settlement during our campaigns. However, our campaigns consisted of continuous movement through varying regions, and therefore, long term stationary deployment in highly particulate waters may potentially lead to settlement. Overall the set-up showed a good performance with continuous data collection providing values within the expected ranges for $p$CO$_2$ across different salinity areas and when split into lakes rivers and channels (Hope et al., 1996; Bouillon et al., 2007; Lynch et al., 2010). However, in comparison to rivers and streams of similar size, $p$CH$_4$ determined in this study had generally higher overall concentrations (Wang et al., 2009; Crawford et al., 2017) and higher overall medians (Stanley et al., 2016). Yet, they are within the range found for other freshwater systems, and on a similar scale with other regions showing large increases in CH$_4$ concentrations (Bange et al., 2019). When focusing on the discrete sample comparison between the calculated $p$CO$_2$ (from TA and DIC) and measured $p$CO$_2$ (HC-CO$_2$) in the limnic cruise (section 3.3.1), the deviation was not unexpected due to the likely presence of organic alkalinity that causes an unknown TA bias that leads to an offset in the calculated $p$CO$_2$ (Abril et al., 2015).

Having the combination of all these sensors, especially with CH$_4$, makes this set-up more unique for measurements across the LOAC. Due to the needed high accuracy for oceanic $p$CO$_2$ measurements, optimisation and continuous improvements of these measurements has been occurring for decades (Körtzinger et al., 1996; Dickson et al., 2007; Pierrot et al., 2009), yet for the comparatively narrow range of oceanic conditions. Sensors have undergone multiple developments and improvements over these years, with the focus for measurements within these water bodies on high accuracy for a relatively small concentration range. In the market currently, there are few oceanic $p$CO$_2$ sensors capable to measure under environmental conditions crossing the boundaries from limnic to oceanic, including the SAMI-CO$_2$ Ocean CO$_2$ Sensor (Sunburst Sensors, LLC, USA, see DeGranpre et al., 1995, Baehr and DeGrandpre, 2004 and Phillips et al., 2015). However, with similar accuracies in the ocean, one advantage of the continuous NDIR/TDLAS based instruments used here, is that no chemical consumables are required for the measurements (refer to review papers for discussion of different sensors: Clarke et al., 2017 and Martz et al., 2015, and for current technological updates on carbonate chemistry instrumentation, refer to The International Ocean Carbon Coordination Project, www.ioccp.org). Traditional flow-through systems on the other hand, such as the commonly used GO system (General Oceanics Inc., Miami, USA), are generally larger, more complex, and built from more components. They also require more maintenance (c.f. reference gases) and the data acquisition is therefore more labour-intensive, also increasing the probability for human error. Sensors designed for inland water bodies tend to be on the lower price range for various reasons, unfortunately leading to lower accuracies and greater inconsistencies (Meinson et al., 2016; Friedlingstein et al., 2019; Canning, 2020). Measurements across the LOAC need high accuracy sensors as the concentrations and dynamic ranges usually decrease from inland

waters to the ocean, and thus, have to match with the oceanic standards, and the set up presented here was designed to fulfil these requirements.

Due to higher quantity and quality, temporal and spatial measurements needed (Natchimuthu et al., 2017), below we present data examples from our various field campaigns illustrating the utility and observational power of our approach to resolve both spatial and temporal variability in parallel for all measured quantities and at very high resolution.

## 420   4.1  Temporal variability

In the Danube delta, portability of the set-up allowed to focus on temporal variability for specific regions over 3 seasons. Due to small power consumption, the small generator and car batteries were sufficient to easily run the entire set-up, allowing for high-resolution (up to 1 Hz), continuous measurements to extract diel cycles in the same way over the 3 seasons. Figure 11 displays data from a two-week field campaign (Rom3), with areas of stationary measurements over extended time periods (grey

shading in Fig. 11, Fig. 3c for locations). Data were continuously logged for all parameters throughout the campaigns, with interruptions only when the houseboat docked. During each campaign, temporal variability showed differences between the regions (lakes, rivers and channels). The stationary phase of the first campaign was in a channel next to lake Isac (Fig. 11, grey box on 16.10., duration 15 h:26 min, Fig. 3c1). An instant peak in $CO_2$ and $CH_4$ can be seen when entering the channel from the lake, coinciding with a drop in $O_2$. Over this diel cycle, $CO_2$ and $O_2$ are apparently governed by production and respiration,

as to be expected (Nimick et al., 2011), yet with relatively constant and high $CH_4$ concentrations. However, during the second stationary zone measurements (Fig. 11, 19.10, Fig. 3c2) conducted within a lake, $pCO_2$ is shown to increase steadily during the station, while always remaining far lower than within the channel. The same diel pattern is shown in the final station stationary phase (Fig. 11, 23.10, Fig. 3c4), which is located in one of the northern channels, far from any lake. These comparisons (from channel to lake variabilities) throughout the transects show the temporal variabilities within regions adjacent, or within close

proximity to one another. Differing vastly in both magnitude and diel pattern, even when comparing the same region (channel next to a lake, 16.10, and a further northern channel, 23.10 Fig. 11, Fig. 3c4).

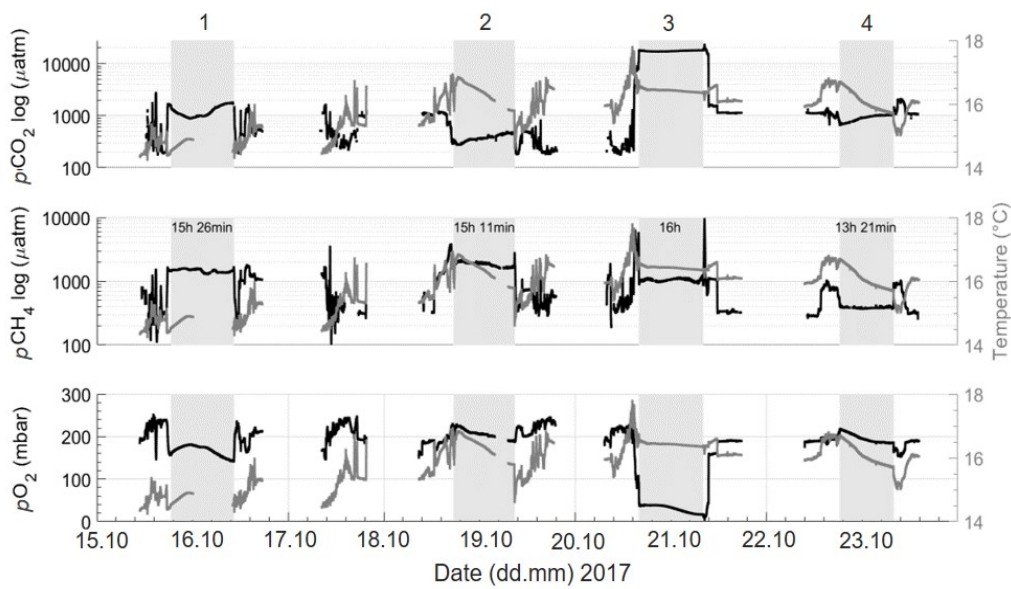

**Figure 11.** Sections acquired during the autumn limnic cruise: Rom3, showing pCO$_2$ ($\mu$atm, logarithmic scale), pCH$_4$ ($\mu$atm, logarithmic scale) and pO$_2$ (in mbar) and temperature ($^{\circ}$C) (grey) from the SBE, across the entire deployment. Temperature is kept constant on the right y-axis for direct changes to be noticed within each gas. Shaded areas and numbers (1 – 4) indicate periods of stationary observations when anchored in one location (see Fig. 3c 1 – 4), with station-keeping durations in hours and minutes given in the middle row. Gaps in data collection refer to the systems being switched off.

Looking closer into specific temporal variabilities, Fig. 12 shows an exemplary 24 h-cycle within a small channel. This location was marked as a 'hot spot' within our transect, showing drastic concentration changes with clear coupling between O$_2$, $p$CO$_2$ and temperature. The $p$CO$_2$ increases from 5,000 $\mu$atm to nearly 17,000 $\mu$atm over the night, then decreases back to initial levels during the day, coinciding with sunrise and sunset, while the opposite trend for both temperature and $p$O$_2$ was observed. Timing and amplitude of these diel trends could have been lost with discrete sampling alone. Due to the same diel variation observed from this location over two of the three months (Rom1 and Rom2), uncertainties behind this variation, such as passing of water parcels anomalies or wind-driven variation as suggested before (Serra and Colomer, 2007; Van de Bogert et al., 2012), can be ruled out as possible explanation. Although diel cycles in inland waters have been investigated (for channels, estuarine, lakes and pond investigations respectively, see: Nimick et al., 2011; Maher et al., 2015; Andersen et al., 2017; van Bergen et al., 2019; Canning et al., 2020), they are generally left out when it comes to average concentrations and corresponding fluxes. Evaluating our data gives evidence that such practices have to be critically evaluated, especially given the abundance and magnitude of diel cycles observed in these regions. Furthermore, allowing for multiple gases to be measured simultaneously, enables extreme observations, such as this, to shed some light on the processes involved (Canning et al., 2020). Therefore, any study aiming to measure representative concentrations and fluxes for limnic systems with significant diel variability, will have

to address this. Adequate sampling/observation schemes should be implemented to avoid strong biases (e.g. by both day-night sampling or by convoluting spatial and temporal variability in 24 h non-stationary mapping exercises).

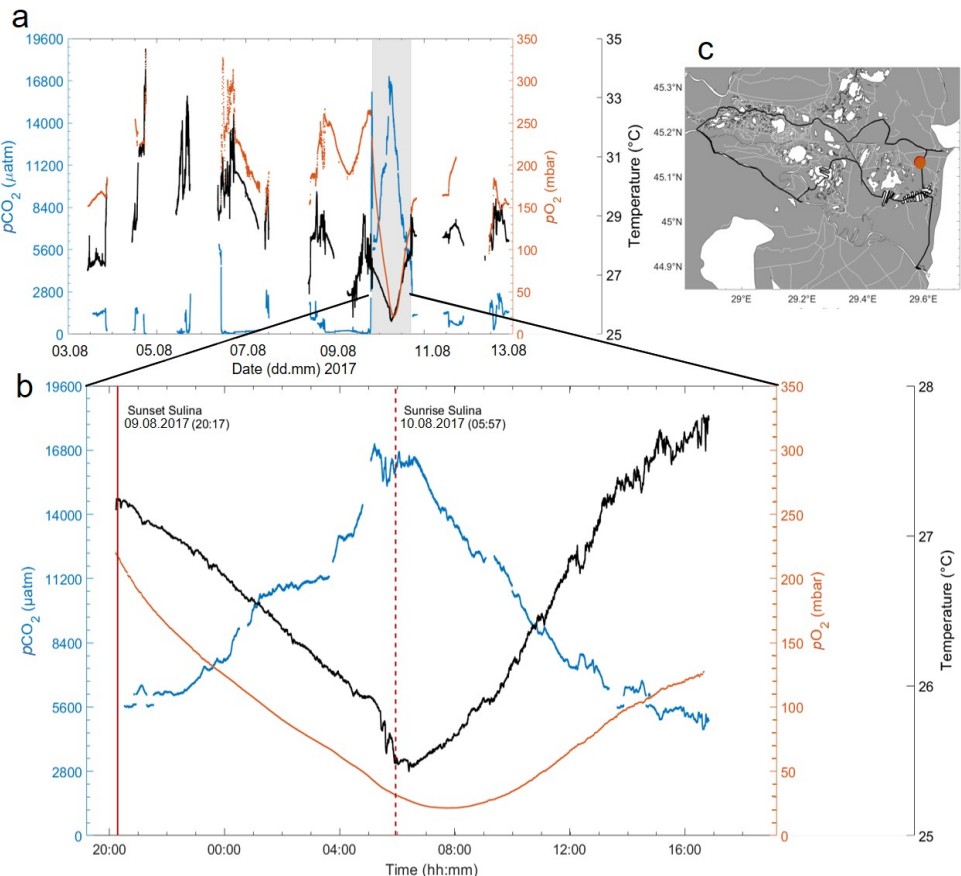

**Figure 12.** (a) Measurements of $pCO_2$ (in micro atm, blue), $pO_2$ (in mbar, orange) and temperature (in °C, black), during the Danube river campaign Rom2 in summer 2017. The grey rectangle highlights a 24 h cycle acquired at a fixed location in a channel; (b) Close up of this 24 h cycle with sunset (solid red), sunrise (dashed line red) indicated, showing extreme variability on the diel cycle time scale.(c) Cruise track of the campaign, with the red dot indicating the position of the 24h stationaly data data aquisition.

## 4.2 Spatial variability

During the limnic campaigns, $CH_4$ showed extreme spatial and temporal differences which highlights the need for high spa-
tiotempral coverage. Although, RT-Corr is not a new method within the ocean/brackish waters (see e.g. Fiedler et al., 2013; Gülzow et al., 2011; Miloshevich et al., 2004), the results of both the HC-CO$_2$ and HC-CH$_4$ corrections show high promises and absolute need in freshwaters for such sensors measuring in highly spatially diverse regions. Both system stability and sensitivity could be demonstrated during the oceanic cruise (15.12.2016 – 13.01.2017: Fig. 13). The little spatial variability was

expected over the large distance when crossing the open ocean waters of the South Atlantic Gyre. The fact that even these small variations in $pCO_2$, $O_2$, and temperature still show clear correlations, points at the very low noise level of the measurements. The Brazil Current and Malvinas Current merge when entering the Patagonian shelf, creating upwelling with fresh nutrients and therefore strongly increased primary production (Matano et al., 2010). These waters are characterized by high productivity with higher $pO_2$ and lower $pCO_2$ (Fig. 13) and increased overall variability compared to the open ocean. Some of these variabilities show the dynamic mixing between the contrasting waters masses of the confluent surface currents. This region is one of the most productive and energetic regions throughout the ocean and is generally poorly described within models due to such dynamics (Arruda et al., 2015). The area is therefore an ideal location to demonstrate the high spatio-temporal resolution of our continuous, automatic multi-parameter approach.

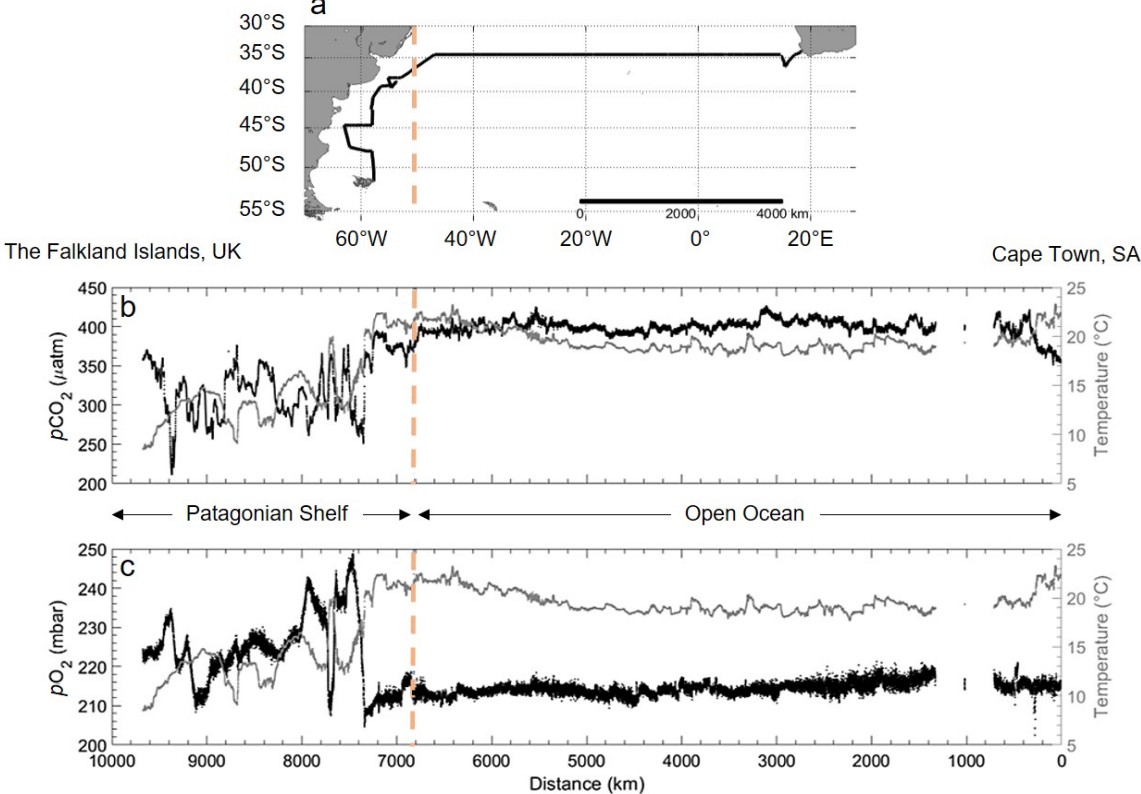

**Figure 13.** Data from the RV Meteor cruise M133 across the Atlantic Ocean (Cape Town/South Africa to the Falkland Islands/UK). a) map of the cruise track (black line) across the Atlantic, orange line showing start of the cruise track entered the Patagoian Shelf, b) $pCO_2$ (in $\mu$atm, black points) and c) $pO_2$ (in mbar, black points) with water temperature (in °C, grey). Note the inverted x-axis to coincide with the direction on the map. Indication of the Patagonian Shelf area (left of the orange dashed line) and open ocean (right of the orange dashed line).

In contrast to the utility for reliably mapping vast ocean regions, Rom1-3 enabled to observe very small-scale spatial variability. Channels were noticeably playing a part in the spatial distribution of high $pCO_2$ and $pCH_4$ throughout the Danube

Delta, as for most freshwater areas (Crawford et al., 2017). This is clearly observed during mapping of the St George river branch (Fig. 14). Although the variability of $pCO_2$ is relatively small ($\sim \pm 60\ \mu$atm), the higher concentrations are still picked up and observed to originate from a side channel, dispersing down the river branch. Although expected, due to the real-time measurement visualization by the CONTROS Detect software, spatial impacts from the channels within more sensitive regions were immediately noticed, allowing for data-guided mapping. The versatility enabled us to complete small spatial scale tran-

sects, with repetitions over time to ensure the concentration changes were primarily due to spatial and not temporal variability (see multiple transects in Fig. 14). This enables spatial dispersion distances to be measured on such small scales.

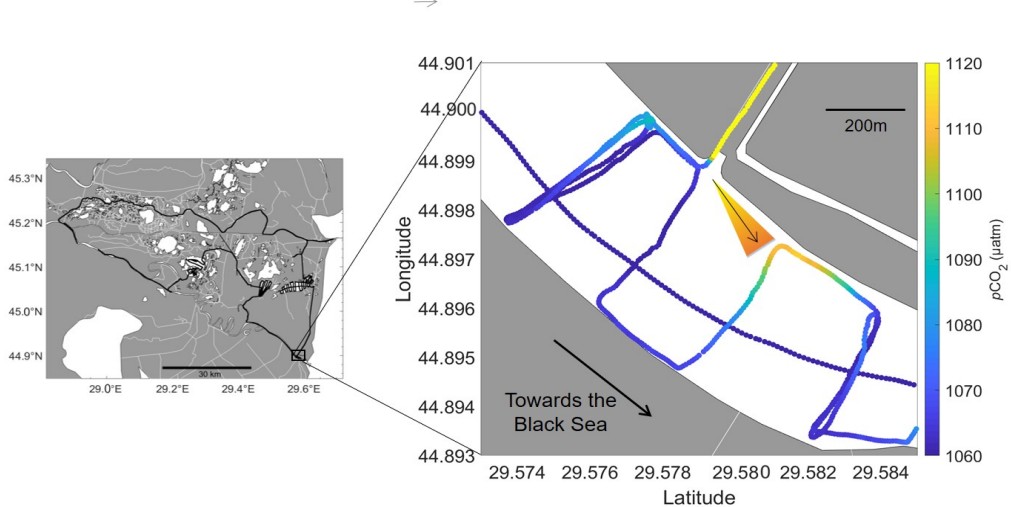

**Figure 14.** Small-scale spatial variability in $pCO_2$ ($\mu$atm) recorded in the Danube Delta from our river transects next to the entrance of a channel near St. George. Direction of the water flow was visible even with small changes in concentrations (arrow and interpretation of concentration gradient and flow direction is indicated to support interpretation).

In more extreme cases, small-scale spatial changes were observed in areas of joining channels during Rom3, where the $pCO_2$ values decreased from 14,722 $\mu$atm to 1,623 $\mu$atm in just over 4 min (Fig. 15). With the house boat travelling between 2-3 knots this corresponds to a distance of about 400 m (Fig. 15). This change was detected within a manmade channel joining Lake

Roşulette towards the Sulina River Branch, arriving from the highest $pCO_2$ and $pCH_4$, along with the lowest $O_2$ throughout the delta transect ($pCO_2$ indicated on the map in Fig. 15). Also shown in Fig. 15 are the processed as well as the raw output from the HC-$CO_2$ (orange dashed), exemplifying the need for all corrections and post-processing steps described above to fully reveal the true spatial distribution. 'Hot spots' and areas of spatially extreme dynamics could be easily passed over with discrete or intermediate sampling. Therefore, this ability to gather such data allows for better classification of individual systems.

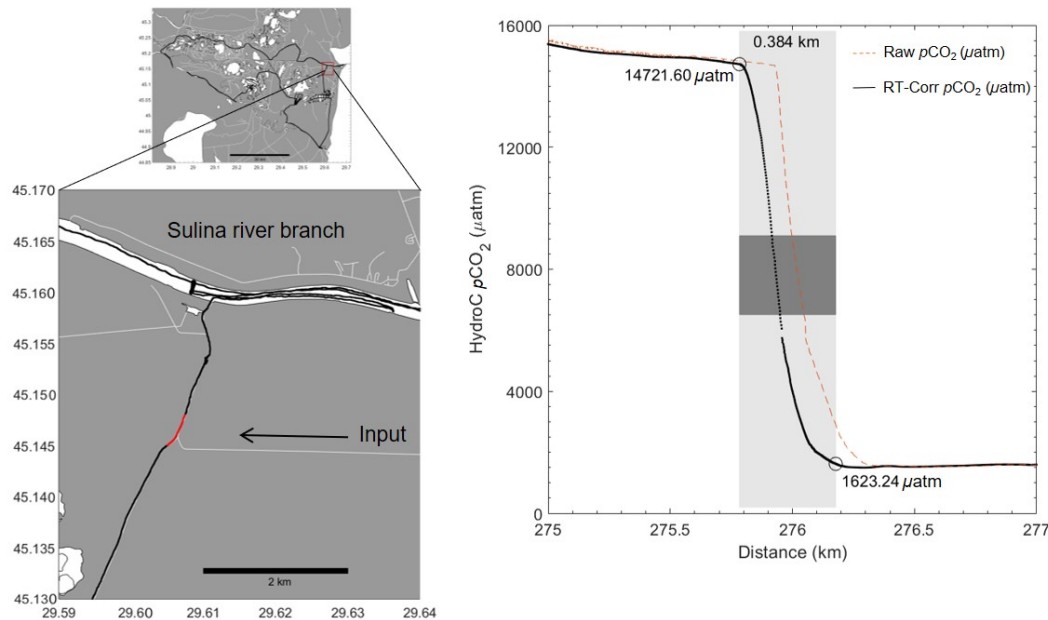

**Figure 15.** Extreme $pCO_2$ concentration gradient over a short time period ($\sim$ 4 min indicated by the red line on the map (left), and light grey box on graph (right)) during Rom3. Raw $pCO_2$ (orange dashed), compared with post-processed, response time corrected (RT-Corr) $pCO_2$ data (black), improving the spatial allocation of the gradient region by $\sim$ 100 m. The gradient occurred over a distance of about 400 meters due to another channel providing a different water source (white lines indicating channel). The dark grey box symbolizes the area over the concentration change in which the houseboat passed the entrance of the entering channel.

Although not shown here, even concentration fluctuations due to vessels passing by were picked up immediately within the data, usually leading to increasing $CO_2$ and $CH_4$ concentrations. With recreational activities and boat usage within some regimes increasing, this should be considered when measuring both fluxes and overall concentrations.

### 4.3   Limitations and benefts

    As we have shown this set-up can be used in the most variable of environments, picking up small variabilities and allowing for
meter by meter readings. The benefit of the system of being built for oceanic precision on the lower concentrations, allows for the system to be continuously used over the boundaries of the LOAC. However, limitations of the system, such as the potential power supply for certain deployments and the long response time (of which has been shown to be overcome by application of the RT-Correction), are outweighed by the benefits of such a system: relativelty long-term stability, reduced demand of human effort required compared to other systems, being able to pick up small variabilities with the response time correction (see
section 3.2). The system allows continuous measurements in combination of other parameters across all salinities, and has a precision and accuray acceptable for measurements in oceanic waters.

# 5 Conclusions

As one of the few studies to combine a sensor package across the entire LOAC for $CO_2$, $CH_4$ and $O_2$ measurements, the importance of seamless observation across the entire LOAC is becoming more apparent. Enabling and openly assessing a variety of techniques across all water types is essential to improve our understanding of carbon budgets and processes especially within the inland regions. We have therefore tried to introduce oceanic precision and attention to detail into the field observations in inland water regions, to potentially allow for measurements in regions of little to no data with a relatively cheap, fully enclosed, sensor package with oceanic accuracy.

The results clearly demonstrate the observational power this technology can provide, but at the same time, illustrate the need for dedicated data processing addressing sensor issues (e.g. drift, calibration range, time constants) for achievement of high data quality. Although all corrections were important, the RT-Corr for $pCH_4$ was viewed as vital when measuring in such a diverse regime (in inland waters), and therefore such practices should be applied. The extended calibration laboratory experiments showed the ability to access higher concentration data values. Despite a slight increase of the error margin, these methods allow for accessing such high values with these sensors, while keeping the precision of the lower concentrations. The results from the suite of campaigns presented here provide further evidence that techniques and sensors designed for specific regimes, can be adapted and when carefully assessed, provide precise measurements across boundaries and through highly diverse regions. Proving oceanic sensors can be used across salinities in a portable way, with little attention needed during operation.

Improvements can be made in terms of size, individual placement of the sensors and accessibility, however, this setup and data readings show the vitality of having high spatiotemporal resolution multi-gas data for mapping and diel cycle extraction, which can further assist with modelling efforts and assessing concentrations and fluxes (Canning et al., 2020). Given there is much need for both high spatial data coverage and accurate concentrations for inland $CO_2$ and $CH_4$ measurements (Crawford et al., 2014; Meinson et al., 2016; Yoon et al., 2016; Natchimuthu et al., 2017; Grinham et al., 2018), this type of data can help fill the gap in this specific region and/or mixing regimes. This can enable better classification of regions, thus furthering monitoring activities and overall carbon budget investigations, which benefit from enhanced data acquisitions on higher spatial and temporal resolutions.

The main use of this continuous, high resolution data can be split into four main sections: (a) large scale monitoring and mapping efforts, (b) temporal variability observations (i.e., with observations in a fixed location or in Lagrangian perspective), (c) spatial variability observation (with a moving platform, often resulting in a convolution of spatial and temporal variability) and (d) the assessment of the coupling between the different continuously observed parameters. The use of separate techniques from oceanography to limnology are slowly becoming unnecessary but there is a definite need for standardized corrections and postprocessing in limnology, such as in the ocean.

*Data availability.* All data is being uploaded to PANGAEA. Canning, Anna; Körtzinger, Arne; Fietzek, Peer; Rehder, Gregor (2020): High-resolution sensor data for $p$CO$_2$, O$_2$ and temperature/salinity from METEOR M133: Cape Town to the Falklands. PANGAEA, https://doi.org/10.1594/PANGAEA.925069.

*Author contributions.* Anna Canning, Arne Körtzinger and Peer Fietzek discussed and designed the study together. Anna Canning collected and processed the sensor data and measured the discrete samples for alkalinity, dissolved inorganic carbon (excluding for M133) and methane, as well as processed the sensor data. Peer Fietzek assisted with processing of this sensor data. Gregor Rehder provided the reference system data for the Baltic Cruise. All authors reviewed the manuscript.

*Competing interests.* All authors declare there is no conflict of interest.

*Acknowledgements.* The research leading to these results has received funding from the European Union's Horizon 2020 research and innovation program under the Marie Sklodowska-Curie grant agreement No 643052 (C-CASCADES project). We thank the captain and crews of the R/V Elizabeth Mann Borgese, R/V Meteor, R/V Littorina and from the Romanian cruises. We would also like to thank Marie-Sophie Maier, Bernhard Wahrli and Christian Teodoru, ETH Zürich, for the project collaboration and Romanian cruise assistance. Also to
540 thank all of the KMCON/-4H-JENA team with their support and help throughout and Matthias Zimmerman, Eawag Kastanienbaum, for the development of the boat logger. Finally, we also thank those who helped with either laboratory or processing assistance; Björn Fiedler, Tobias Steinhoff, Tobias Hahn, Katharina Seelmann, Alexander Zavarsky, Dennis Booge and Jennifer Clarke, all originally from GEOMAR, Kiel.

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
