# Peer review of "Technical Note: Seamless gas measurements across Land-Ocean Aquatic Continuum - corrections and evaluation of sensor data for $CO_2$ , $CH_4$ and $O_2$ from field deployments in contrasting environments"

_Biogeosciences, 2020_

## Referee Comment (RC1) · Anonymous Referee #1 · 25 May 2020

In this paper the authors combine off the shelf sensors for pCO2, pCH4, pO2 temperature and salinity into a flow through system and assess the utility of the system for measuring spatio-temporal variability of these parameters across the land-ocean interface.

Overall I found the paper to be well written and clearly presented. I have a few suggestion for improvement:

[Figure]

1. To me, the pCH4 system with an apparent offset from standard methods, as well as an extremely long response time does not sit well within the stated aims of developing a system capable of detecting spatio-temporal variations across the land ocean boundary. Can the authors expand upon this, perhaps the pCH4 system described is advantageous for some experiments, but not so for others. Good to be upfront with the limitations as well as highlighting the benefits.

2. As the paper is currently written, it is hard to see what the advantage of the proposed system over the traditional methods for measuring these parameters. The description of calibrations, RT offsets etc are really useful, but I think a section dedicated to benefits over currently available systems would add value. This could cover aspects like power consumption, size, cost etc. For pCO2 - equilibrator-NDIR systems can cover an equally large concentration range, are cheaper, and have an equally quick RT. So essentially I am left asking, what are the benefits of this system for pCO2 measurements? Same for pCH4, although sensor cost is higher than CO2 NDIR and RT is also long (although quicker than reported for the Contros system presented here).

3. While the e-folding method of assessing RT is commonly used, I think it would also be useful to highlight the t90 values. This gives the reader a more relevant and directly relevant understanding of RT without having to do additional calculations to assess actual RT.

A few typos Ref page 2 ln 39 change PA to Raymond

Ln 281 The comment on no biofouling- later in the paper you highlight that biofouling may be responsible for some of the discrepancies (e.g. ln 317). Best to keep the message consistant

Ln 396 What is meant by "abnormal peaks", perhaps rephrase

Ln 384 - "from point of stationary", while I understand what is meant here, the terminology is a bit clunky, perhaps rephrase.

---

## Referee Comment (RC2) · Mariana Ribas-Ribas (Referee) · 25 Aug 2020

**General comments**

I like the paper a lot, especially the detailed data of day-night, the "lost with discrete sampling alone" and of short spatial resolution. I congratulate the first author as I think is graduate student and I appreciate the sampling effort and technology preparation.

My General comments are:

- Maybe more to the Editor than to the authors. Key point of this paper is this link between limnologist and oceanographers (l. 34). As far as I can say, all authors, editor and myself are in the oceanography site. If the other reviewer is too, it might be good to have a limnologist reading it. I know it is difficult, more in this times, but worth to check it otherwise it will again be a bias and we are again only talking to the same people.
- I understand and agree about the differences methods between ocean and lakes but I think measurement in the Baltic Sea use same methods as oceanic and have the same quality. Just need to rewrite some of sentence when refers to it, because as it reads now I look three complete different "words" and I would say we have two
- General organization might be improved, for example there is quite a lot of discussion in the result section (lines 283-284, lines 295-298…) and results on the discussion (l. 279-384). From figure 11 on, there have been not presented in the result section.
- Statistics. Figure 10 both data sets (oxygen measured and Hydroflash included uncertainty, there is not one dependent controlled and the other one independent) and therefore a model II regression should be used (see for example Legendre, 2014)
- I am not sure what the journal policy but I recommend all data to be available via PANGAEA.

**Specific comments and technical corrections**

L. 9 Define what NDIR and TDLAS are.

L. 23 what is TS? Probably reference editor Problem.

L. 37 something not right with this sentence "due to using" (?)

L. 46 there are more studies reporting high pCO2 values, for example in Guadalquivir River (Ribas-Ribas et al., 2011).

L. 60-61 Again something not right with this sentence and the , used in between.

L. 75 define NDIR

L.80 I'm not completely sure you do not discuss biogeochemical implications. I am also not sure why not to do it.

L. 82 is water flow an ancillary data?

L. 95 ( in wrong place

L. 100 what is the frequency for the other measurements?

Figure 1. I do not understand how do you define the flow. What is the 9 referring to?

Table 1. You are showing conductivity, not salinity. In the RT, what is 1:32 and 22:46? Careful with number-space-unit (5 L min in pCO2 RT) and subscript in 63.

L. 118 such as ??

L. 124 Fig. 2, Table 2 and section Number

Figure 3. Unify how you refer to Falklands in all the figures. In Fig. 3c it will be nice to have the details on where you did the diurnal cycles and all other features you describe in the test.

L. 129 German date style (different from the figures, unify). Also in line 153, table 2

L. 135. Where was the SBE21 located?

L. 134 What happen to the other half of the brackish cruise. Did you improve the method on the spot? Same question I have in line 150.

L. 152 Probably a question to an English native speaker but "Excursions" does not sound good.

L. 154-157 Add all this details to Fig. 3b.

L. 158 Were the discrete samples collected before or after the sensor? Will pump/flow modify the carbonate chemistry?

L. 163 I do not understand why there were not SBE data… the system is design to measure in parallel, right?

L. 164 use same decimal places for the standard deviation

L. 168 100 mL (capital L, you did good with the 500 mL ☺)

Table 2. Vessel size would not need decimal places. Confusing how the ranges are reported, consider to add a ":" like in the header or "-"

L. 172-174 So now I am curios, were there differences? But more important for the paper, if they are not reported or need it, you do not need to say it ☺

L. 182 VINDTA and APOLLO had difference accuracies and precisions, report both. See (Tynan et al., 2016)

L. 183-185. What about the other constant used?

L. 192 First define and the (MESS)

L. 195 Again here, if only one is reported, why mention?

L. 198-199 Here and elsewhere in the ms, unify the units of methane and $CO_2$ (sometimes ppm, some other µatm). I think this is another thing oceanographers and limnologist do different ☺. Also in line 229, 238

L. 205 Is RT already mention and defined?

Section 2.5 I like this section a lot. I think it will benefit to separate the corrections in subheaders: 1 (or 2.5.1)) sensor drift; 2) warming…

L. 218 Explain better what info the shape of the zero drift give to you.

L. 224 Define SST and the depth of this "surface"

Equation 1 and 2. Unify how do you write insitu.

Line 228 at in situ ?? (I guess temperature but need to be stated)

Line 295 Replace believe for think

Line 303-305. Figure 5 b is discussed before 5a

Figure 6. If I only read the captions (which a lot of readers do) I do not know what it is the true. I would also change the word, how do you know it is true? Maybe reference

Line 317 A few lines before you say biofouling was not an issue.

Figure 7 miss a), b), c)… Also the caption is not clear enough. Maybe too many information for only one figure (?)

Line 326. You have filtered and unfiltered samples to know if this TA bias was from that.

Figure 8 also miss a) and b). Do not use left and right, or bottom (in line 345)

Figure 9 c x-axis miss (dd/mm/year). Unify throughout the figures (also figure 11 is missing it). Also it is really difficult to see the difference in the lower scale. Consider to do a break in the y-axis

Figure 10. how do you re-calibrated, with which data?

Line 359 long deployment is a bit relatively, compare with other really long deployment (more than a year ☺)

L. 384-388 This info will be better if state in the map of the study area. Where the stations on grey in Figure 11 are in the map in figure 3c

Figure 11. Why show three times the same temperature. Consider to add and extra subplot.

End of section 4.1 is excellent, probably the main part of the paper. You mention that diel cycles in inland are scare. I will say that we do not know a lot on what happen during night. We recently published a paper that open more questions than answers (Stolle et al., 2020).

Figure 12. Same comments about a), b) and date format

L 409 Strange sentence with the verb between ","

Figure 13. Transect if from Cape Town to Falkand but these two place are in different panels and it is somehow confusing. It seems like a relatively straight line, could you use longitude instead of distance

Figure 14. Where is St. George. Unify decimal places in all axis (44.900 and no 44.9). Also unify how you report Latitude and longitude (map is different from transect). Miss a), b)… Same for figure 15

Line 445-446 "it gives light to one way to access" (?)

References I haven't go through all of them but please pay attention to reference format. Examples: line 492 East China Sea (with strange spaces) and line 512 (Control Procedures…, all capital)

Ribas-Ribas, M., Gómez-Parra, A., and Forja, J. M.: Air–sea $CO_2$ fluxes in the north-eastern shelf of the Gulf of Cádiz (southwest Iberian Peninsula), Mar. Chem., 123, 56-66, 2011.
Stolle, C., Ribas-Ribas, M., Badewien, T. H., Barnes, J., Carpenter, L. J., Chance, R., Damgaard, L. R., Quesada, A. M. D., Engel, A., Frka, S., Galgani, L., Gašparović, B., Gerriets, M., Mustaffa, N. I. H., Herrmann, H., Kallajoki, L., Pereira, R., Radach, F., Revsbech, N. P., Rickard, P., Saint, A., Salter, M., Striebel, M., Triesch, N., Uher, G., Upstill-Goddard, R. C., Pinxteren, M. v., Zäncker, B., Zieger, P., and Wurl, O.: The MILAN Campaign: Studying Diel Light Effects on the Air–Sea Interface, B Am Meteorol Soc, 101, E146-E166, 10.1175/bams-d-17-0329.1, 2020.

Tynan, E., Clarke, J. S., Humphreys, M. P., Ribas-Ribas, M., Esposito, M., Rérolle, V. M. C., Schlosser, C., Thorpe, S. E., Tyrrell, T., and Achterberg, E. P.: Physical and biogeochemical controls on the variability in surface pH and calcium carbonate saturation states in the Atlantic sectors of the Arctic and Southern Oceans, Deep Sea Research Part II: Topical Studies in Oceanography, 127, 7-27, https://doi.org/10.1016/j.dsr2.2016.01.001, 2016.

---

## Author Comment (AC1) · 15 Sep 2020

**Responses to Anonymous Referee #1**

**bg-2020-128**

In this paper the authors combine off the shelf sensors for pCO2, pCH4, pO2 temperature and salinity into a flow through system and assess the utility of the system for measuring spatio-temporal variability of these parameters across the land-ocean interface.

Overall I found the paper to be well written and clearly presented. I have a few suggestion for improvement:

Response: Thank you for these comments!

1. To me, the pCH4 system with an apparent offset from standard methods, as well as an extremely long response time does not sit well within the stated aims of developing a system capable of detecting spatio-temporal variations across the land ocean boundary. Can the authors expand upon this, perhaps the pCH4 system described is advantageous for some experiments, but not so for others. Good to be upfront with the limitations as well as highlighting the benefits.

Response: Thank you for your comment. Concerning the offset from standard methods, as stated within the manuscript, the $p$CH$_4$ system accuracy is ± 2 µatm or 3% of the reading. For the brackish water campaign the sensor is within these specifications. For the limnic campaigns data are not within the 3% of the reading some of the time, as judged from comparison with discrete. We note and state in the manuscript that in a situation of high variability matching of underway data with discrete samples, one can have high matching uncertainty (specific lines 339-348) leading to apparent offsets which are partly incurred by inappropriate matching. It must also be noted that determination of dissolved CH$_4$ concentrations from discrete samples is also not fully mature yet, and have shown significant inter-laboratory offsets (Wilson et al., 2018; stated in the manuscript). We show in this study that with mathematical corrections however, the drawback of a long response time can be overcome. Please see line 307 ff. for very successful RT correction, where the corrected data vary in tight anti-correlation with the pattern of the O$_2$ data which have ~3 second response time. In addition to longer-term station deployments, where fast response time is not needed due to the slow change in $p$CH$_4$ concentration, we can therefore demonstrate the sensor's applicability in highly variable environments (Canning et al,. In Prep).

Of course, this system does have limitations, however when focusing on the advantages – long term stability or being able to pick up small variabilities with the response time correction (see section 3.2) while having continuous measurements in combination of other parameters – we believe this out-ways the limitations. We believe we have discussed and been open with limitations, however to make this clearer, a review of the manuscript has resulted in sentences becoming sharper based on this comment to ensure both the limitations and benefits have been presented.

Canning, A., Wehrli, B., and Körtzinger, A., 2020. Methane in the Danube Delta: the importance of spatial patterns and diel cycles for atmospheric emission estimates. *Biogeosciences Discussions* (soon to be submitted)

2.  As the paper is currently written, it is hard to see what the advantage of the proposed system over the traditional methods for measuring these parameters. The description of calibrations, RT offsets etc are really useful, but I think a section dedicated to benefits over currently available systems would add value. This could cover aspects like power consumption, size, cost etc. For pCO2 - equilibrator-NDIR systems can cover an equally large concentration range, are cheaper, and have an equally quick RT. So essentially I am left asking, what are the benefits of this system for pCO2 measurements? Same for pCH4, although sensor cost is higher than CO2 NDIR and RT is also long (although quicker than reported for the Contros system presented here).

Response: Thank you for your comment. Within the paper the advantages (and limitations) of the system we feel have been addressed. The purpose was to combine and have multiple fully autonomous oceanic sensors to be able to measure across the boundaries, simultaneously. This can enable both oceanic and also limnic waters to be measured with the same system. Although there are separate sensors out there, these do not fulfill having all the necessary characteristics for both the ocean and inland waters: a system that measures all 3 gases completely autonomously, able to cope with a range of outside temperatures and variable conditions, non-demanding in terms of equilibrator systems and calibration gases, with only one person needed to supervise and conduct measurements in more demanding environments, while measuring simultaneously and being able to deal with the high accuracy needed in the ocean (of which these are tested, and are currently used in the ocean). The aim was all of these, while combined with being able to then also measure steep changes and high concentrations for all 3 gases. In terms of oceanic sensor prices, these are on the lower end, and portable enough to be adapted to inland water sampling. Therefore, specifically for $pCO_2$ measurements, we believe we showed the benefits of this system are stated above as we found it was able to cope in such challenging environments, yet could handle the accuracy needed, while measuring as a combined system. However, we can see your concern and sentences have been added into the manuscript with other systems for comparison and allow the reader to know what else is out there and therefore addressing the advantages and benefits of this system.

3.  While the e-folding method of assessing RT is commonly used, I think it would also be useful to highlight the t90 values. This gives the reader a more relevant and directly relevant understanding of RT without having to do additional calculations to assess actual RT.

Response: As we recommend $t_{63}$ and correct for it later in the paper, we have just added $t_{90}$ data in for comparison when discussing the response time correction. For the HC $CO_2$ it would be 212s and for the HC $CH_4$ it would be 3145s.

A few typos Ref page 2 ln 39 change PA to Raymond

Response:  This has been corrected, thank you.

Ln 281 The comment on no biofouling- later in the paper you highlight that biofouling may be responsible for some of the discrepancies (e.g. ln 317). Best to keep the message consistent

Response: On line 281, this is for no biofouling on the membrane itself, for line 317 it is for potential biofouling within the tubing, which was significantly longer for that specific cruise. I have changed it to make this clearer.

Ln 396 What is meant by "abnormal peaks", perhaps rephrase

Response: (actually line 369) Abnormal peaks is meant by small specific regions of increased concentration. This is rephrased to '…other regions showing large increases in $CH_4$ concentrations.'

Ln 384 - "from point of stationary", while I understand what is meant here, the terminology is a bit clunky, perhaps rephrase.

Response: Rephrased to 'However, during the second stationary zone measurements (Fig.11, 19/10) conducted within a lake, $pCO_2$ is shown to increase steadily during the station keeping while always remaining far lower than within the channel.'

---

## Author Comment (AC2) · 15 Sep 2020

**Responses to Mariana Ribas-Ribas, Referee #2**

**bg-2020-128**

I like the paper a lot, especially the detailed data of day-night, the "lost with discrete sampling alone" and of short spatial resolution. I congratulate the first author as I think is graduate student and I appreciate the sampling effort and technology preparation.

Response: Thank you!

My General comments are:

- Maybe more to the Editor than to the authors. Key point of this paper is this link between limnologist and oceanographers (l. 34). As far as I can say, all authors, editor and myself are in the oceanography site. If the other reviewer is too, it might be good to have a limnologist reading it. I know it is difficult, more in this times, but worth to check it otherwise it will again be a bias and we are again only talking to the same people.

Response: The work was done in collaboration with limnologists from ETH Zürich and Eawag, Switzerland (Marie-Sophie Maier, Bernhard Wehrli and Christian Teodoru). Although they were not specifically involved in the processing, we either worked in parallel for the collection of data, and/or are working on further analyses of the data for manuscripts in preparation. They are therefore aware of this manuscript and the content. Bernhard Wehrli has also reviewed a previous version of this paper within the PhD thesis chapter of the first author. They are mentioned and thanked for their contributions in the acknowledgments.

Bernhard Wehrli:  https://usys.ethz.ch/en/people/profile.bernhard-wehrli.html

- I understand and agree about the differences methods between ocean and lakes but I think measurement in the Baltic Sea use same methods as oceanic and have the same quality. Just need to rewrite some of sentence when refers to it, because as it reads now I look three complete different "words" and I would say we have two

Response: I have changed this, although the idea was to show it can go across the entire salinity range. When addressing "marine" and "brackish" (i.e., $0.5 < S < 30$) as a different "world" we refer to the fact that many chemical methods need special adaptation to this low-salinity range (e.g., alkalinity titration).

- General organization might be improved, for example there is quite a lot of discussion in the result section (lines 283-284, lines 295-298…) and results on the discussion (l. 279-384). From figure 11 on, there have been not presented in the result section.

Response: Thank for the valuable comment. We will revise the paper accordingly.

- Statistics. Figure 10 both data sets (oxygen measured and Hydroflash included uncertainty, there is not one dependent controlled and the other one independent) and therefore a model II regression should be used (see for example Legendre, 2014)

Response: Although one is not controlled, the Winkler oxygen measurements methods have been well tested. However, we ran also a model II regression giving an $R^2$ of 0.988 (original 0.97). This will be replaced.

- I am not sure what the journal policy but I recommend all data to be available via PANGAEA.

Response: The data will be uploaded asap!

Specific comments and technical corrections

ns L. 9 Define what NDIR and TDLAS are.

Response: NDIR is a nondispersive infrared sensor and TDLAS is Tunable diode laser absorption spectroscopy, this has been added to the manuscript.

L. 23 what is TS? Probably reference editor Problem.

Response: Thanks for spotting this error in the reference. We have corrected this.

L. 37 something not right with this sentence "due to using" (?)

Response: Rephrased to 'Often, this is due to different measuring techniques and protocols, both with respect to in-situ/autonomous observations and the collection of discrete data.'

L. 46 there are more studies reporting high pCO2 values, for example in Guadalquivir River (RibasRibas et al., 2011).

Response: Thank you, extra citations have been inserted, including this one.

L. 60-61 Again something not right with this sentence and the , used in between.

Response: Rephrased to 'These methods often focus on only one gas (usually $CO_2$) and none of these methods mentioned, cover both waters types. On top of this, spatiotemporal data coverage has been noted to be sparse and is needed to advance our budget accuracies and understanding.'

L. 75 define NDIR

Response: This is already defined on line 58.

L.80 I'm not completely sure you do not discuss biogeochemical implications. I am also not sure why not to do it.

Response: I do not go into detail here, more due to length and specifics of the manuscript. To go into full detail here of biogeochemical implications and analyses would lose the sense of this manuscript. A further 2 manuscripts have been prepared from this alone which will be mentioned in the manuscript.

L. 82 is water flow an ancillary data?

Response: For the corrections but not the actual data. Water flow has been separated from the 'ancillary data'.

L. 95 ( in wrong place

Response: We have changed this.

L. 100 what is the frequency for the other measurements?

Response: Added in frequencies of all the sensors.

Figure 1. I do not understand how do you define the flow. What is the 9 referring to?

Response: It is defined as L/min, 9 is referring to the total flow (9 L/min). We have added this in.

Table 1. You are showing conductivity, not salinity. In the RT, what is 1:32 and 22:46? Careful with number-space-unit (5 L min in pCO2 RT) and subscript in 63.

Response: Corrected this, thank you! The 1:32 and 22:46 are in minutes, on the following line it says this. We have made this clearer.

L. 118 such as ??

Response: This was not meant to have the 'and' there, it has been corrected.

L. 124 Fig. 2, Table 2 and section Number

Response: We have corrected this to section number.

Figure 3. Unify how you refer to Falklands in all the figures. In Fig. 3c it will be nice to have the details on where you did the diurnal cycles and all other features you describe in the test.

Response: Some are described with the specific examples, such as in figure 12 with the 24 hour cycles, but we have added in the others!

L. 129 German date style (different from the figures, unify). Also in line 153, table 2

Response: This has been corrected, thank you.

L. 135. Where was the SBE21 located?

Response: The SBE21 was located within the mess room, and the water inlet was located on the bulbous bow a few meters below the water level. This has been included in the manuscript.

L. 134 What happen to the other half of the brackish cruise. Did you improve the method on the spot? Same question I have in line 150.

Response: There was an internal issue of the detector related to absorption peak identification which is stated in line 137. There was nothing that could be done at the time, it was later fixed when back on land by the manufacturer.

L. 152 Probably a question to an English native speaker but "Excursions" does not sound good.

Response: It is perfectly fine to use in our opinion, but it can be changed to campaigns for enhanced clarity.

L. 154-157 Add all this details to Fig. 3b.

Response: These details have been added in.

L. 158 Were the discrete samples collected before or after the sensor? Will pump/flow modify the carbonate chemistry?

Response: They were collected before the sensors from a little tube, connected to the tubing of the sensor package with a valve. As flow was recorded, in terms of the corrections it didn't do any difference, and also the amount of water used to fill the samples was minimal and therefore didn't have much influence (given 9L/min water flow). The way we set it up, it didn't create any bubbles either into the system and for each sample stage we saw no effect when taking the samples given the high resolution. Each time we took a sample it was usually when stationary, and following this we would stay in the same place for a while and as we were able to observe the data immediately, we saw no obvious influence.

L. 163 I do not understand why there were not SBE data… the system is design to measure in parallel, right?

Response: Yes, however these were 2 very specific points when we experienced power-cuts overnight, and therefore not picked up immediately by us. All of the sensors, except for the SBE unit, restarted automatically. For following campaigns (not shown in this manuscript) this issue was solved.

L. 164 use same decimal places for the standard deviation

Response: This has been corrected, thank you.

L. 168 100 mL (capital L, you did good with the 500 mL ☺)

Response: This has been corrected, thank you.

Table 2. Vessel size would not need decimal places. Confusing how the ranges are reported, consider to add a ":" like in the header or "-"

Response: This has been made clearer, thank you.

L. 172-174 So now I am curios, were there differences? But more important for the paper, if they are not reported or need it, you do not need to say it ☺

Response: Yes, there were slight differences, however this is not reported here as it is out of the specifications of this manuscript. It has been removed so not create confusion.

L. 182 VINDTA and APOLLO had difference accuracies and precisions, report both. See (Tynan et al., 2016)

Response: These have been put in the manuscript.

L. 183-185. What about the other constant used?

Response: For pH scale and $KSO_4$ dissociation constants: seawater and Dickson & TB of Uppstrom 1979, this is added into the paper too.

L. 192 First define and the (MESS)

Response: This has been corrected.

L. 195 Again here, if only one is reported, why mention?

Response: This has been removed, as stated above.

L. 198-199 Here and elsewhere in the ms, unify the units of methane and CO2 (sometimes ppm, some other µatm). I think this is another thing oceanographers and limnologist do different ☺. Also in line 229, 238

Response: Thanks for spotting the inconsistency in line 229, where indeed µatm is the correct unit. For the other two examples (l. 198-199 and l. 238) it's fine to mention that the sensors were calibrated with standard gases characterized by certain $CO_2$ concentrations in ppm. We have revised and clarified the text by adding that even if calibrated with $xCO_2$ final data by the sensors was converted to $pCO_2$ in µatm.

L. 205 Is RT already mention and defined?

Response: Yes, it is already mentioned on line 109.

Section 2.5 I like this section a lot. I think it will benefit to separate the corrections in subheaders: 1 (or 2.5.1)) sensor drift; 2) warming…

Response: Thank you! We will do this.

L. 218 Explain better what info the shape of the zero drift give to you.

Response: This is worded incorrectly, thank you. The "shape of the zero drift" was not considered, we considered the temporal change in the concentration-dependent response of the sensor between pre- and the post-cruise factory calibration, i.e. span drift, during processing to be linear to the sensor's runtime. This sentence has been corrected to say this.

L. 224 Define SST and the depth of this "surface"

Response: We have included this.

Equation 1 and 2. Unify how do you write insitu.

Response: The insitu had been unified!

Line 228 at in situ ?? (I guess temperature but need to be stated)

Response: Yes, this has been inserted.

Line 295 Replace believe for think

Response: We have done this.

Line 303-305. Figure 5 b is discussed before 5a

Response: This has been rearranged to make it in chronological order.

Figure 6. If I only read the captions (which a lot of readers do) I do not know what it is the true. I would also change the word, how do you know it is true? Maybe reference

Response: We have changed this. Instead of "true", we have changed this to 'inverted $pO_2$ mbar (grey) as a technically independent yet parameter-wise linked reference for 'real-time' spatiotemporal variability, i.e. RT $O_2$ sensor << RT $CH_4$ sensor'

Line 317 A few lines before you say biofouling was not an issue.

Response: 'no clogging or biofouling of the membranes' on line 281, is for no biofouling on the membrane itself. On line 317 where it says, 'possibly pointing at the onset of a biofouling issue within the tubing', is for potential biofouling within the tubing, of which was significantly longer for that specific cruise. We have changed this to make it far clearer.

Figure 7 miss a), b), c)… Also the caption is not clear enough. Maybe too many information for only one figure (?)

Response: We have made this clearer!

Line 326. You have filtered and unfiltered samples to know if this TA bias was from that.

Response: Unfortunately, we did not collect filtered and unfiltered samples for this. We collected poisoned and un-poisoned (not used), which is the reason why this data is not included. We have removed any mention of this as it would just create confusion!

Figure 8 also miss a) and b). Do not use left and right, or bottom (in line 345)

Response: This is been changed.

Figure 9 c x-axis miss (dd/mm/year). Unify throughout the figures (also figure 11 is missing it). Also it is really difficult to see the difference in the lower scale. Consider to do a break in the y-axis

Response: This has been unified and made clearer!

**Figure 10. how do you re-calibrated, with which data?**

Response: With the linear offset found with the discrete samples. The re-calibration comes from the offset found between the sensor and discrete samples. Significant optode sensor drift, particularly when the sensor is not in the water, is a well-documented phenomenon (Bittig et al., 2018). This has been added to the MS: 'an average offset of -19.04 ± 2.26 (± RMSE) µmol L$^{-1}$ and -29.78 ± 5.04 (± RMSE) µmol L$^{-1}$ was also found within the open ocean and shelf respectively to discrete oxygen data. Two separate linear offset corrections were applied throughout all oceanic data.'

Bittig, H.C., Körtzinger, A., Neill, C., van Ooijen, E., Plant, J.N., Hahn, J., Johnson, K.S., Yang, B. and Emerson, S.R., 2018. Oxygen optode sensors: principle, characterization, calibration, and application in the ocean. *Frontiers in Marine Science*, 4, p.429.

Line 359 long deployment is a bit relatively, compare with other really long deployment (more than a year ⍰)

Response: We have made this clearer and put various campaigns lengths instead.

L. 384-388 This info will be better if state in the map of the study area. Where the stations on grey in Figure 11 are in the map in figure 3c

Response: This would have to involve making the figure far larger. A separate figure has been made for this to make all points clear.

Figure 11. Why show three times the same temperature. Consider to add and extra subplot. Response: This was designed as such so you could see the changes clearly over the diel cycles instead of looking down and skipping details. We have written this in the caption that it this is the same temperature.

End of section 4.1 is excellent, probably the main part of the paper. You mention that diel cycles in inland are scare. I will say that we do not know a lot on what happen during night. We recently published a paper that open more questions than answers (Stolle et al., 2020).

Response: Thank you! We agree, and thank you for the matching reference, it will be included in the manuscript.

Figure 12. Same comments about a), b) and date format

Response: This has been changed.

L 409 Strange sentence with the verb between ","

Response: This has been rephrased slightly

Figure 13. Transect if from Cape Town to Falkand but these two place are in different panels and it is somehow confusing. It seems like a relatively straight line, could you use longitude instead of distance

Response: This wouldn't work with the transect going south on the Patagonian shelf (the left panel). We will find a better solution and make both panels more consistent!

Figure 14. Where is St. George. Unify decimal places in all axis (44.900 and no 44.9). Also unify how you report Latitude and longitude (map is different from transect). Miss a), b)... Same for figure 15

Response: We have unified this and added in a)... into the figure.

Line 445-446 "it gives light to one way to access" (?)

Response: This has been rephrased to 'Although increasing the error, these methods allow for accessing such high values with these sensors, while keeping the precision of the lower concentrations.'

References I haven't go through all of them but please pay attention to reference format. Examples: line 492 East China Sea (with strange spaces) and line 512 (Control Procedures..., all capital)

Response: This has been sorted out.

Ribas-Ribas, M., Gómez-Parra, A., and Forja, J. M.: Air–sea CO2 fluxes in the north-eastern shelf of the Gulf of Cádiz (southwest Iberian Peninsula), Mar. Chem., 123, 56-66, 2011. Stolle, C., Ribas-Ribas, M., Badewien, T. H., Barnes, J., Carpenter, L. J., Chance, R., Damgaard, L. R., Quesada, A. M. D., Engel, A., Frka, S., Galgani, L., Gašparović, B., Gerriets, M., Mustaffa, N. I. H., Herrmann, H., Kallajoki, L., Pereira, R., Radach, F., Revsbech, N. P., Rickard, P., Saint, A., Salter, M., Striebel, M., Triesch, N., Uher, G., Upstill-Goddard, R. C., Pinxteren, M. v., Zäncker, B., Zieger, P., and Wurl, O.: The MILAN Campaign: Studying Diel Light Effects on the Air–Sea Interface, B Am Meteorol Soc, 101, E146-E166, 10.1175/bams-d-17-0329.1, 2020.

Tynan, E., Clarke, J. S., Humphreys, M. P., Ribas-Ribas, M., Esposito, M., Rérolle, V. M. C., Schlosser, C., Thorpe, S. E., Tyrrell, T., and Achterberg, E. P.: Physical and biogeochemical controls on the variability in surface pH and calcium carbonate saturation states in the Atlantic sectors of the Arctic and Southern Oceans, Deep Sea Research Part II: Topical Studies in Oceanography, 127, 7-27, https://doi.org/10.1016/j.dsr2.2016.01.001, 2016

Response: Thank you for the references.

---

## Author Response (AR2)

**Responses to Editor bg-2020-128**

L26 The sentence "Regions such as lakes, rivers and reservoirs, are only more recently becoming recognized as significant elements of the global carbon budget (e.g. Regnier et al. 2013; Borges et al., 2015)" is not fully exact. In fact, $CO_2$ emissions

5   from lakes and rivers have been first recognized as significant more than 20 years ago. In 1995 by Cole et al. 1995 in Science (lakes) and Cole and Caraco 2001 Mar. Freshwater Res., 2001, 52, 101–10 (rivers) and then updated with new (pH/TA) data compilation by Tranvik et al. (2009 Land) and Raymond et al 2013. Nothing really new on the quantification of lakes and rivers $CO_2$ emissions in Regnier's 2013 paper, which does not provide any new data, but rather uses the previously published estimates:

**Response:** Thank you for this point. The thinking behind this sentence was to show it was relatively less (and more recent) compared to the ocean, as in the sentence before I say how it is known they are supersaturated for over 50 years, however I can see this is not so clear. I have changed the sentence, taking in your suggestion.

15   L32 and 75. Sometimes high-resolution measurements as those you are proposing here allow some necessary changes in paradigms on aquatic C cycle, compared to those derived from discrete "blind" sampling (see eg Richey et al. 2002 and Abril et al. 2014 both in Nature). This might be another powerful argument to promote the development of in situ, high resolution and real-time instrumentations.

20   **Response:** Thank you for this comment. I have added in a few points for this!

**Extras:**

Added a legend to Figure 15 for clarity and figure fonts unified

Data avaliability has been updated

[revised manuscript text omitted]